# Polyphasic Systematics of the Fungicolous Genus *Cladobotryum* Based on Morphological, Molecular and Metabolomics Data

**DOI:** 10.3390/jof8080877

**Published:** 2022-08-20

**Authors:** Nikola Milic, Anastasia C. Christinaki, Dimitra Benaki, Aimilia A. Stavrou, Nikolaos Tsafantakis, Nikolas Fokialakis, Vassili N. Kouvelis, Zacharoula Gonou-Zagou

**Affiliations:** 1Section of Ecology and Systematics, Department of Biology, National and Kapodistrian University of Athens, Panepistimiopolis, 157 84 Athens, Greece; 2Section of Pharmacognosy and Chemistry of Natural Products, Department of Pharmacy, National and Kapodistrian University of Athens, Panepistimiopolis, Zografou, 157 71 Athens, Greece; 3Section of Genetics and Biotechnology, Department of Biology, National and Kapodistrian University of Athens, Panepistimiopolis, 157 84 Athens, Greece; 4Section of Pharmaceutical Chemistry, Department of Pharmacy, National and Kapodistrian University of Athens, Panepistimiopolis, Zografou, 157 71 Athens, Greece

**Keywords:** anamorphs, ITS region, metabolomics, morphology, mycophilic fungi, NMR, phylogenetics

## Abstract

(1) Background: Species of the anamorphic genus *Cladobotryum*, are known for their fungicolous lifestyle, making them important mycopathogens in fungiculture. Many morphological, ecological, and molecular phylogenetic studies of the genus have been done to date, but taxonomic uncertainties and challenges still remain. Fungal secondary metabolites, being vastly diverse, are utilised as an extra tool in fungal systematics. Despite being studied for their potentially bioactive compounds, *Cladobotryum* species are insufficiently investigated regarding metabolomics. (2) Methods: The aim of this study is the identification of Greek strains of *Cladobotryum* by integrating morphological data, ITS-based phylogeny, and ^1^H NMR-based metabolomics into a polyphasic approach. (3) Results: Twenty-three strains, isolated from sporophores of macromycetes inhabiting diverse Greek ecosystems, were morphologically identified as *Cladobotryum apiculatum*, *C. fungicola*, *C. mycophilum*, *C. varium*, *C. verticillatum*, and *Hypomyces rosellus* (anamorph *C. dendroides*), whereas seven strains, which produced red-pigmented metabolites, presented an ambiguous taxonomic position at the species level. Molecular phylogenetics and metabolomics corroborated the morphological findings. (4) Conclusions: Thorough morphological study, ITS region-based phylogeny, and NMR-based metabolomics contribute complementarily to resolving the genus *Cladobotryum* systematics.

## 1. Introduction

Species encompassed within the ascomycetous genus, *Cladobotryum* Nees, represent a fraction of anamorphs (conidial or asexual states) of the pleomorphic hypocrealean genus, *Hypomyces* (Fr.) Tul. and C. Tul. (Ascomycota, Pezizomycotina, Sordariomycetes, Hypocreomycetidae, Hypocreales, Hypocreaceae). Teleomorphs (sexual states) of *Cladobotryum* outside the genus *Hypomyces* are not hitherto reported, whereas *Hypomyces* is associated with several other anamorphic genera, some of which are now merged into *Cladobotryum* [1,2]. For many Hypocreales, including the genus *Cladobotryum*, the teleomorph–anamorph interconnections remain unresolved [3,4,5]. However, in some *Cladobotryum* species, the teleomorph was observed in culture—or rarely in nature—and typified as *Hypomyces* [6,7,8,9,10,11,12]. According to the International Nomenclature Code for Algae, Fungi, and Plants (ICN), and following the “one fungus, one name” principle, a priority species name must be defined and established in current use for taxa having names for both morphs [13,14]. There were proposals for the displacement of *Cladobotryum* in favor of *Hypomyces* [3,15], however the matter seems to still be pending. 

Hypocrealean fungi establish a plethora of biological interactions in the environment [4]. However, the ecological hallmark of many Hypocreales—and especially of the genus *Cladobotryum* (and its teleomorph *Hypomyces*)—is a fungicolous (mycetophilous or mycophilic *sensu* Rudakov [16]) lifestyle [2,17,18,19,20]. Fungicolous fungi are characterised by the development of a stable association with other fungi (hosts on which they grow) [20,21]. The host fungi of the genus *Cladobotryum* are mainly macromycetes, principally the sporophores of basidiomycetes, but also those of ascomycetes, and this kind of fungal–fungal interaction appears to vary in terms of host specificity [20,22,23]. Finally, some *Cladobotryum* species are important mycopathogens of cultivated mushrooms, causing serious pathological conditions to basidiomes, called “cobweb disease”, thus having negative implications on fungiculture [24,25,26,27]. On the other hand, this ecological feature opened new horizons for the exploitation of the genus *Cladobotryum* as a potential biocontrol agent against fungal phytopathogens [28]. The micromorphology of the anamorphic state is characterised by (sub)verticillately-branched conidiophores from which subulate conidiogenous cells emerge, producing mainly septate hyaline conidia in various manners, and usually by formation of non-dehiscent, thick-walled resting sclerotioid structures [29]. In the teleomorphic state, mostly two-celled, apiculate ascospores are produced within perithecia [1]. Multiple morphological and ecological studies of the genus *Cladobotryum* (and the teleomorphic *Hypomyces*) have been made to date, giving rise to a convenient categorisation of its species into host-based groups (i.e., agaricicolous, aphyllophoricolous, polyporicolous, and boleticolous), thereby emphasising the preference for specific groups of basidiomycetes [6,30,31,32]. Nevertheless, taxonomic difficulties surrounding the genus *Cladobotryum* are rooted in its micromorphology. As is the case with other hypocrealean anamorphs [1], the pleomorphism that characterises this genus, in addition to overlapping of the taxonomic characters among distinct species and the micromorphological variability, which is often observed within a single species, make its taxonomy a challenging feat. 

An early attempt at the utilisation of chemical characters in the systematics of fungicolous fungi—in particular, the genus *Hypomyces* and its anamorphic states—underlined the production of pigmented secondary metabolites (SMs) and their possible taxonomic value [7]. For instance, skyrin, emodin, rugulosin, bikaverin, rosellisin, and particularly the red-hued metabolite, aurofusarin, have been associated with various species of *Cladobotryum/Hypomyces* [7,33,34]. A potential chemotaxonomic significance of some of those pigmented compounds (e.g., aurofusarin) was also implied from previous phylogenetic studies [2,35,36]. Finally, the biological activity of *Cladobotryum* SMs was also reported, indicating a biotechnological potential of the genus [37,38,39,40,41,42,43,44]. 

Genus *Cladobotryum* is insufficiently studied to date when it comes to molecular genotyping and large-scale biochemical/metabolic phenotyping or metabolomics. Considering the former approach, mainly the ribosomal RNA ITS region (ITS1-5.8S-ITS2) has been used, followed by the rarely used molecular markers, 28S and a combination of *rpb1*, *rpb2*, *tef1*, and FG1093 [2,35,36,45,46]. Regarding the second approach, in fungal chemotaxonomy and metabolomics, various sophisticated chromatographic techniques, often coupled with spectral analysis, are used, thereby contributing to a polyphasic approach in systematics by extracting data from multiple sources, such as morphology, genome, (bio)chemistry [47,48,49,50,51,52,53,54,55,56]. Although NMR spectroscopy was used for the structural characterisation and elucidation of compounds produced by *Cladobotryum/Hypomyces* [38,39,40,41,42,43,44], to the best of our knowledge, the aim of the already published studies implementing this analytical technique has not been targeted to taxonomic investigations of the genus so far, per se. The structure of aurofusarin was elucidated decades ago using NMR from *Fusarium*, *Hypomyces rosellus*, and its anamorphic state, *Cladobotryum dendroides* [57,58,59,60].

*Cladobotryum dendroides* and *C. mycophilum*, and their teleomorphs *H. rosellus* and *Hypomyces odoratus*, accordingly, are the most widely-distributed species in Europe, causing cobweb disease in cultivated and wild mushrooms [61,62,63,64,65]. Species with few reports from Europe are *C. varium* (teleomorph *Hypomyces aurantius*), *C. verticillatum*, *C. apiculatum*, *C. fungicola* (teleomorph *Hypomyces semitranslucens*), and *C. penicillatum* [6,7,8,9,10,22,29,30,31,35,36,45]. The appearance of the species, *C. tenue* and *C. rubrobrunnescens*, seems to be restricted to very few countries, i.e., Germany and Estonia for both, and Spain and Israel for each one, respectively [7,35]. Almost all species records are from Northern Europe, whereas very few and sporadic references are available from Southern Europe. Species delimitation, diversity, phylogeography, and host associations of red-pigmented, temperate, and tropical *Cladobotryum/Hypomyces* are addressed in comprehensive works [2,35,36].

The aim of this study is the identification of Greek strains of the fungicolous genus, *Cladobotryum*, by integrating morphological, molecular phylogenetic, and metabolomic data into a polyphasic systematic approach. Coupling of NMR-based metabolomics with morphology and ITS-based phylogenetics was initially piloted in a preliminary study of Greek *Cladobotryum* spp. [66]. The present paper represents the first in-depth, comprehensive study of the genus *Cladobotryum* in Greece and its fungicolous diversity. The fact that macromycetes (both basidiomycetes and ascomycetes) from diverse Greek habitats have often been encountered as hosts of other fungi is especially motivating and offers a fertile ground for further research. Consequently, this study also represents the country’s first official report of fungal hosts of *Cladobotryum* species. 

## 2. Materials and Methods

### 2.1. Fungal Material and Culture Conditions

The total number of investigated strains of the genus *Cladobotryum* amounts to thirty-two, thirty of which were isolated from various fungal hosts inhabiting diverse Greek habitats in different sites, while the other two strains represent ex-type cultures obtained from Westerdijk Fungal Biodiversity Institute (Utrecht, The Netherlands) as reference and comparison material (Table 1). All specimens and strains are deposited at the Mycetotheca ATHUM in the Dried Specimen Collection and Culture Collection of Fungi of the National and Kapodistrian University of Athens (NKUA, Athens, Greece), respectively. The same ATHUM number is assigned to both the specimens and the cultures.

For the morphological study, the isolates were cultured in Petri dishes (Ø 90 mm) on potato dextrose agar (PDA, BD Difco) and incubated at 23 °C in a natural day/night photoperiod, while for the molecular analysis they were grown in potato dextrose broth and incubated in a shaker (200 rpm), under the same conditions. All strains intended for the metabolomic analysis were grown in triplicates in the same conditions as in the morphological study and sent for analysis on the sixteenth day of growth.

### 2.2. Morphological Study

Both the macro- and micromorphology of the fungal colonies were thoroughly investigated for a period of at least four weeks. Macromorphological characteristics that were taken into consideration include colony obverse colour, texture and growth, and colony reverse colour. Additionally, a colour reaction test—regarded as negative or positive—was conducted by applying potassium hydroxide solution (KOH 3%) on actively growing mycelia. Micromorphology was meticulously studied, with emphasis given to anamorphic reproductive structures (conidiophores, conidiogenous cells, and conidia), vegetative structures (sclerotioid), and, if produced, on teleomorphs (perithecia and ascospores). Where possible, conidiogenesis was also observed and described. Conidia sizes are followed by the mean quotient of length and width (Qm). In the micromorphological study of the reference and the unidentified strains, the slide culture method was also applied [67]. Microphotographs were obtained using an AxioImager A1 (Carl Zeiss AG, Oberkochen, Germany) Differential Interference Contrast (DIC) microscope. Microscopic mounts were stained using phloxine B, lactophenol cotton blue, KOH 3%, or Melzer’s reagent. Conidia were measured using AxioVision Rel. 4.8, Zen 3.2 (blue edition) and 3.4 (blue edition) (Carl Zeiss AG, Oberkochen, Germany) and a statistical analysis of spore measurements was conducted using Microsoft Excel and GraphPad Prism 6.0.1 and 8.0.1 for Windows (GraphPad Software, San Diego, CA, USA). For visualisation of the data obtained from the statistical spore analysis, the RAWGraphs visualisation platform [68] was used. The Inkscape 1.1.2 vector graphics editor [69] was used for the assemblage of the figures, customisation of charts, and illustration of the simplified diagrammatic depictions. 

In order to visualise the data derived from conidial measurements (being one of the crucial taxonomic characters), a customised bubble-type chart was chosen to illustrate three conidial dimensions (mean values of)—conidial lengths, conidial widths, and conidial length/with ratios (Qm). The third variable (Qm) is represented—instead of using usual bubbles or circles—via ellipsoidal shapes (not to scale) at each data point, and each one is approximated by the mean length and width values corresponding to each species/strain and are coloured by applying an automated diverging (two-hue) colour scale. A diverging colour scale is chosen to visualise phenomena going towards two different directions or extremes. That is, if the shape of *Cladobotryum* conidia is approximated to an ellipsis, in the case of our specimens, the conidia diversify from a broader ellipsoidal to a narrower ellipsoidal shape. 

### 2.3. Molecular Phylogenetic Analyses

#### 2.3.1. DNA Extraction and PCR Reaction

Mycelia were collected by vacuum filtration and the total DNA isolation was performed using 50 mg of ground fungal material, as previously described [70]. The isolated DNA samples were used for PCR amplification of the ITS1-5.8S-ITS2 region of the Internal transcribed spacer (ITS) using the previously published primers, 18ITS1 [5′-GTCCCTGCCCTTTGTA-3′] and 28ITS2 [5′-CCTGGTGGTTTCTTTTCC-3′] [71]. 

PCR amplification reactions were performed with a KAPA Taq PCR Kit (KAPA Biosystems, Wilmington, MA, USA) in a PTC-200 Gradient Peltier Thermal Cycler (MJ Research, Waltham, MA, USA), according to the manufacturer’s instructions. The amplification protocol for the ITS region was: 3 min at 95 °C; 35 cycles of 30 s at 95 °C, 60 s at 48 °C, 2 min at 72 °C; and a final extension 5-min incubation at 72 °C. PCR amplicons were purified and cleaned using the PCR cleanup kit (NEB, Monarch PCR and DNA Cleanup Kit). All ITS amplicons were sequenced in both directions and assessed using the program SeqMan of Lasergene Suite 11 (DNASTAR Inc., Madison, WI, USA) [72]. The final sequences were deposited into GenBank. (Acc. Nos. OM993297–OM993326).

#### 2.3.2. Phylogenetic Tree Reconstruction

In addition to the 29 ITS sequences produced in this study, 63 publicly available sequences of species belonging to the genera *Cladobotryum* and *Hypomyces* were used for determining the phylogenetic relationships within and among these genera (Table 2). *Fusarium* sp. ATHUM 10261 (C1032B) was used as an outgroup in this analysis (Acc. No. OM 993315) since it belongs to the order Hypocreales. The alignment was created using the E-INS-i method, as implemented in the multiple sequence alignment program MAFFT [73,74]. Alignment parameters were set to default. 

The phylogenetic tree was constructed using both the Neighbor Joining (NJ) method, through PAUP4 [75], and Bayesian Inference (BI) method, through MrBayes [76]. In the NJ method, the reliability of nodes was evaluated using 10,000 bootstrap iterations. In the BI analysis, the most suitable evolutionary model was determined using the program, JmodelTest (ver. 2.0), and it was GTR + G + I, when the BIC Information Criterion was applied [77,78]. For the BI approach, 4 independent MCMCMC analyses were performed, using a sampling set adjustment for every 100,000 generations with a total of 10 million generations. The resulting phylogenetic tree was edited in FigTree v1.4.3 (http://tree.bio.ed.ac.uk/software/figtree/, accessed on 28 July 2022).

### 2.4. NMR-Based Metabolomics

#### 2.4.1. Metabolite Extraction

The experimental design presented in this study was based on previously suggested protocols with some modifications [79,80]. The aim was to capture both endo- and exometabolome of the fungi under study, therefore pieces of each culture (mycelium together with agar medium underneath) were used in the extraction process.

Three pieces of a specified size (approximately 7 mm in diameter) of each 16-day-old culture triplicate underwent quenching by adding cold methanol (−20 °C), which accompanied ultrasound-assisted extraction in a solvent system created by adding an equal volume of ethyl acetate (MeOH:EtOAc, 2:2 mL) for one hour. The quenching/extraction step was conducted in a separate test tube for each triplicate. The obtained extracts were filtrated and concentrated with Rotavapor^®^ R-215 (BÜCHI Labortechnik AG, Flawil, Switzerland). The dried extracts were subsequently re-dissolved in a solvent system of methanol, chloroform, and water (MeOH:CHCl_3_:H_2_O, 1:1:0.9 mL). The re-dissolved extracts were vortexed and then separated into polar (hydroalcoholic, MeOH/H_2_O) and non-polar (chloroformic, CHCl_3_) phases via centrifugation (Thermo Fisher Scientific, Waltham, MA, USA) at 4300 rpm. Afterwards, the polar phases were dried using a nitrogen evaporator REACTI-THERM III #TS-18824 Heating Module (Thermo Fisher Scientific, Waltham, MA, USA) and the extracts were kept at −80 °C until NMR analysis. The sample preparation step for the ^1^H NMR spectroscopy required the re-dissolution of the extracts in deuterated methanol (MeOD) containing tetramethylsilane (TMS) as an internal standard for the calibration of the chemical shift axis. The protocol is depicted in Appendix A.

#### 2.4.2. NMR Spectroscopy

The ^1^H NMR spectra of the strains were acquired on a BRUKER Avance III 600 MHz Spectrometer (BRUKER, Karlsruhe, Germany) equipped with a PABBI z-gradient probe and an automatic sample changer of 60 holders (B-ACS 60). A fully automated procedure including insertion of the sample in the magnet, temperature stabilization (300 K), field homogeneity optimization, pulse calibration, spectra acquisition, Fourier transformation, phase correction, and chemical shift axis calibration were accomplished by IconNMR software based on TopSpin v. 2.1. For each sample, a ^1^H 1D NMR spectrum with water suppression using a combination of 1D NOE pulse sequence with presaturation (noesygppr1D, Bruker library) and a J-resolved pseudo-2D spectrum were recorded. The ^1^H 1D NMR spectra were recorded for a spectral width of 20 ppm with 64k data points, resulting in a 2.66 s acquisition time, 128 repetitions, and a mixing time of 100 ms. J-resolved spectra were recorded with 12k data points over a spectral width of 16 ppm in the ^1^H axis and 40 points in the indirect dimension of 78.12 Hz width.

#### 2.4.3. Statistical Analysis

High-resolution ^1^H 1D NMR spectra, from 9.60 to 0.04 ppm, were segmented into 0.02 ppm-width buckets using the software package, AMIX Statistics v3.9.14 (Analysis of MIXtures, Bruker BioSpinGmbH, Karlsruhe, Germany), and the corresponding integrals were exported as an excel worksheet in order to be subjected to statistical analysis. Solvent trace regions were excluded from the analysis (residual water signal: 4.92–4.76 ppm and methanol signal and satellites: 3.36–3.28, 3.44–3.40, and 3.20–3.18 ppm, respectively) and data were normalised to total intensity by assigning the unit to the sum of the spectrum buckets. A multivariate analysis was performed in the SIMCA v. 14.1 (Umetrics, Umeå, Sweden) environment, which aimed to unravel hidden cluster formations. Principal component analysis (PCA) and the supervised projection to latent structure discriminant analysis (PLS-DA) were applied and the generated models were evaluated from the explained and the predicted fractions of the total data variation. Additionally, a permutation test of 100 random changes were applied for the validation of the PLS-DA models.

## 3. Results

### 3.1. Morphological Study

From the morphological study of the 30 strains, 23 were assigned to the following six species viz. *C. fungicola*, *C. apiculatum*, *C. verticillatum*, *C. varium*, *C. mycophilum*, and *H. rosellus* (anamorph *C. dendroides*), while the remaining 7 could not be identified at the species level. For each of the six identified species and the two reference strains grown on PDA, colony obverse and reverse observations, conidial data, and short comments are given below. Analytical descriptions of these species can be found in Appendix A. Furthermore, all seven unidentified strains are described in detail and are grouped as “Unidentified Red-Pigmented” Strains (URPs).

***Cladobotryum fungicola*** (G.R.W. Arnold) Rogerson and Samuels, Mycologia 85 (2): 262 (1993)

*Strain examined:* Greece: *Fthiotida:* Mesochori, Mt. Oiti, in *Quercus frainetto*, *Q. pubescens*, and *Q. coccifera* forest with *Abies cephalonica* (sporadically), on basidiome of *Cortinarius* sp., 2008, coll./isol. *Z. Gonou-Zagou*, ATHUM 6855.

*Mycelium*: floccose to cottony; white, with prominent tufts; relatively symmetrically spreading, with irregular margin (Figure 1a). *Colony reverse*: pale buff to yellow, progressively turning intensely yellow to bright orange (Figure 1b). *Conidia*: ellipsoidal to cylindrical or fusiform; (13)15–18(20) × 5–6.5 μm, Qm 2.8; mainly two-celled, rarely one-celled; sometimes slightly curved or swollen at base or intensely constricted at septa; hyaline, smooth (Figure 1e–h).

*Comments*: The micromorphological characters of our strain fit perfectly with the original description of the anamorphic state [11], where the species is described as *Sibirina fungicola*. The long, zigzag-shaped apical part of conidiogenous cells is prominent and characteristic (Figure 1c,d). They are vaguely illustrated by Põldmaa and Samuels [31], though they are not mentioned in the text or in the original description.

Hosts of *Cladobotryum fungicola* recorded by now in the available literature belong to various genera of basidiomycetes, mainly aphyllophoraceous [22]. As far as our knowledge goes, this is the first record of the genus *Cortinarius* as its host.

***Cladobotryum apiculatum*** (Tubaki) W. Gams and Hooz., Persoonia 6 (1): 97 (1970)

*Strain examined:* Greece: *Karditsa:* Ag. Nikolaos, Mt. Zigourolivado, in *Fagus sylvatica* forest, on basidiome of *Russula* sp., 2009, coll. *P. Delivorias*, isol. *A. Liakouri*, ATHUM 6907.

*Mycelium:* cottony, compactly to aerial; white to off-white; symmetrically spreading, with regular margin (Figure 2a). *Colony reverse:* white, progressively turning off-white, buff to ochre; sometimes visible brownish granules corresponding to sclerotioid structures (Figure 2b). *Conidia*: ovoid, ellipsoidal to broadly cylindrical or clavate; (11)13–28(33) × (3)5–9.5(11) μm, Qm 3; one-celled, very rarely two-celled; quite often swollen at base; hyaline, smooth; many easily germinating from both edges (Figure 2e,f). *Sclerotioid structures*: present (Figure 2g,h).

***Cladobotryum verticillatum*** (Link) S. Hughes, Canadian Journal of Botany 36 (6): 750 (1958).

*Strains examined:* Greece: *Arkadia:* Megalopoli, Mt. Lykaio, in *Quercus* sp. forest, on basidiome of *Lactarius subumbonatus*, 2009, coll. *M. Triantafyllou*, isol. *Z. Gonou-Zagou*, ATHUM 6850; Megalopoli, Mt. Lykaio, in *Quercus* sp. forest, on basidiome of *Lactarius subumbonatus*, 2010, coll. *M. Triantafyllou*, isol. *A. Liakouri* ATHUM 6920; Megalopoli, Mt. Lykaio, in *Quercus* sp. forest, on basidiome of *Lactarius subumbonatus*, 2010, coll. *M. Triantafyllou*, isol. *A. Liakouri* ATHUM 6921.

*Mycelium:* felt-like to cottony; white to off-white; symmetrically spreading, with regular margin (Figure 3a). *Colony reverse:* white, progressively turning off-white to very pale buff (Figure 3b). *Conidia:* subglobose to ovoid or broadly ellipsoidal; 11–24(27) × (6)8–11(13) μm, Qm 2.2; one-celled; extremely rarely two-celled; hyaline, smooth; many easily germinating from both edges; few joined at their base or laterally (Figure 3e–f). *Sclerotioid structures*: present (Figure 3g,h).

*Comments:* Our strains of *C. apiculatum* and *C. verticillatum* are quite similar macromorphologically in culture. In terms of micromorphology, the most notable difference between the two species is the presence of mainly one-celled, mostly subglobose to ovoid conidia in *C. verticillatum*, in contrast to one- to rarely two-celled, ovoid to broadly elongated conidia in *C. apiculatum*. The measurements of the micromorphological characters fit with the various descriptions [8,29,30].

Both species have a preference to russuloid basidiomes, as expected [22]. *Cladobotryum apiculatum* occurs on *Russula* sp. and all *C. verticillatum* strains were isolated from the basidiomycete *Lactarius subumbonatus*, which is reported, to the best of our knowledge, for the first time as a host of the genus *Cladobotryum* [22].

***Cladobotryum varium*** Nees, System der Pilze und Schwämme: 56, t. 4:54 (1817)

*Strains examined:* Greece: *Attiki:* Mt. Parnitha, in *Abies cephalonica* forest, on basidiome of *Clitocybula familia*, 2008, coll./isol. *Z. Gonou-Zagou*, ATHUM 6845; Mt. Parnitha, in *Abies cephalonica* forest, on basidiome of *Cortinarius* sp., 2009, coll./isol. *Z. Gonou-Zagou*, ATHUM 6856; Mt. Parnitha, in *Abies cephalonica* forest, on basidiome of *Panellus* sp., 2009, coll./isol. *Z. Gonou-Zagou*, ATHUM 6846; *Eurytania:* near Krikello, in *Abies boresii-regis* forest, on basidiome of *Inocybe* sp., 1998, coll./isol. *Z. Gonou-Zagou*, ATHUM 6514; near Krikello, in *Abies borisii-regis* forest, on polypore basidiome, 1998, coll./isol. *Z. Gonou-Zagou*, ATHUM 8003; Mt. Tymfristos, in *Abies borisii-regis* forest with *Juniperus oxycedrus* (sporadically), on basidiome of *Ganoderma* sp., 2004, coll./isol. *Z. Gonou-Zagou*, ATHUM 8002; *Karditsa:* Ag. Nikolaos, Mt. Zigourolivado, in *Fagus sylvatica* forest, on agaricoid basidiome, 2009, coll. *P. Delivorias*, isol. *A. Liakouri* ATHUM 6908; *Rodopi:* Frakto virgin forest on polypore basidiome, 2000, coll. *Th. Angelopoulos*, isol. *Z. Gonou-Zagou* ATHUM 7996; *No data:* on polypore hymenophore, 1997, coll./isol. *Z. Gonou-Zagou*, ATHUM 7995.

*Mycelium:* slightly felt-like or cottony to powdery or with floccose to powdery tufts radially or irregularly arranged; white, sometimes yellow at centre; usually when mature with bright yellow-orange irregular patches; symmetrically spreading, with regular margin (Figure 4a). *Colony reverse:* buff, progressing towards various yellow hues with ochre and orange undertones (Figure 4b). *Conidia:* ellipsoidal to shorty cylindrical; (10)13–17(20) × (5)6–8(9.5) μm, Qm 2.1; mainly two-celled, rarely one-celled or three-celled; sometimes constricted at septa; base with small apiculus or truncate, sometimes with a ring-like formation; many produced in a slanting manner; hyaline, smooth; predominantly single, sometimes forming chains, often branched (Figure 4g,h). *Sclerotioid structures*: present (Figure 4i).

*Comments:* Our isolates exhibit inter- and intrastrain variability. Nevertheless, the colonies of *C. varium* are distinguished macromorphologically from those of all other studied species by their remaining white, powdery mycelium; the irregular orange patches, usually formed on their surface—a feature not referred to in the existing descriptions—and the buff-yellow-orange reverse. Microscopically, the conidia are mostly ellipsoidal and two-celled, and in the strains, ATHUM 6846 and 4856, form characteristic long chains, some of which branched. This uncommon feature was also observed by Matsushima in only one strain (MFC 1816 = CBS 675.77) [81] and was referred to by Rogerson and Samuels [6]. Conidial connections via anastomoses in the strain, *C. varium* ATHUM 6846, were also observed [82]. In addition, the heterogeneity of conidiogenous cells is apparent in our strains, especially in colonies of ATHUM 8002 and ATHUM 6908, where the cells, when maturing, become gradually shorter and from their truncated apices two types of conidia are gradually produced—ellipsoidal conidia truncated at the base and cylindrical conidia with both sides truncated—revealing a probable retrogressive mode of conidiogenesis (Figure 4d–f). This characteristic ontogeny of the conidia is described by Cole and Kendrick [83]. In any case, the conidiogenesis observed is similar only to that of *C. mycophilum*.

The basidiomycetes *Clitocybula familia*, *Ganoderma* sp., *Inocybe* sp., and *Panellus* sp., from which our strains were isolated, as far as our knowledge goes, are not reported as hosts of *C. varium* [22].

***Cladobotryum mycophilum*** (Oudem.) W. Gams and Hooz., Persoonia 6 (1): 102 (1970)

*Strains examined:* Greece: Attiki: cultivated *Agaricus bisporus*, 2010, coll./isol. *Z. Gonou-Zagou*, ATHUM 8001. *Eurytania:* Ag. Nikolaos, in *Platanus orientalis* forest with *Quercus* sp., *Abies cephalonica* and *Castanea sativa* (sporadically), on basidiome of *Mycena* sp., 2010, coll./isol. *Z. Gonou-Zagou*, ATHUM 8000; *Karditsa:* Belakomitis, in *Abies borisii-regis* forest, on basidiome of *Inocybe* sp., 1999, coll./isol. *Z. Gonou-Zagou*, ATHUM 7994. *Magnisia:* Mt. Pilio, in *Castanea* sp. forest, on basidiome of *Hypholoma* sp., 2009, coll. *P. Delivorias*, isol. *A. Liakouri* ATHUM 6906.

*Mycelium*: felt-like at first, becoming fibrous, floccose to cottony; white, yellow, buff to ochre, progressing towards various red hues with brownish and purplish undertones; when mature with white powdery tufts, aerial or superficial, irregularly or radially arranged; symmetrically or asymmetrically spreading, with regular margin; sometimes distinct, minute brownish granular formations; rarely production of perithecia (Figure 5a). *Colony reverse*: yellow, buff, progressing towards ochre, pinkish hues, and various red hues with brownish undertones (Figure 5b). *Conidia*: ovoid to broadly ellipsoidal to ellipsoidal or cylindrical; mainly two-celled, many three-celled, few four-celled, and rarely one-celled; some curved; variably sized, two-celled (16)20–28(32) × (7)9–11.5(13), Qm’ 2.1, 3-4-celled 25–35(39) × 9–12 μm, Qm″ 3.1, Qm = (Qm′ + Qm″)/2 = 2.6; hyaline, smooth; at base with apiculi sometimes indistinct or truncate; characteristic production in a slanting manner indicative of retrogressive conidiogenesis; mature conidia with hyaline pore-like protrusions, some joined in twos or forming short chains; production of secondary conidia on a conidiogenous cell emerging directly from a conidium (Figure 5g–k). *Sclerotioid structures*: present (Figure 5l). *Teleomorph:* only in strain ATHUM 8001; *Perithecia:* mostly subglobose to pyriform, reddish to purplish brown; on ostiole either slightly pigmented droplets or light yellowish mass of ascospores; partially immersed in the subiculum (Figure 5m). *Ascospores:* fusiform to oblong, apiculate; 25–30(32) × 4–5.5(6.5) μm (incl. apiculi); two-celled, sometimes with constriction at median septum; many curved; apiculi rather conical, acute to rarely obtuse; 4.5–5.5(6.5) μm; slightly verrucose (Figure 5n).

*Comments*: All strains studied exhibited intra- and interstrain variability as far as macro- and micromorphology is concerned—i.e., colony texture (including formations such as anamorphic sporulation, sclerotioid aggregations, ascigerous structures) and colour and production of pigmented metabolites, and their corresponding microscopic characters. Nevertheless, our descriptions—and the flunctuations of the characters—fit perfectly with the protologue of *Hypomyces odoratus* [10], where both sexual and asexual reproduction were described from culture. According to our observations, the anamorph is characterised by mainly two-celled conidia, but also three- to four- celled specimens, varying in size and shape, which are produced retrogressively from gradually shortening conidiogenous cells. Conidiogenesis in *C. mycophilum* is similar to that of *C. varium* and very characteristic of the retrogressive way of production. It is worth noting that secondary conidia seem to be produced from mature conidia and most probably the hyaline protrusions of the latter indicate the place of their blastic formation. Joined conidia were depicted in a drawing by Arnold [10], though this trait was not mentioned in the text. Sexual reproduction was observed in the strain, ATHUM 8001, which was isolated from a cultivated basidioma of *Agaricus bisporus*. Perithecia were formed in the very first cultures, with the strain losing this ability with time. The teleomorphs seem to be produced only in culture and are described very few times [8,9,10,46], while the fungus appears as an anamorph in nature. A species of the genus *Hypholoma*, from which one of the studied strains was isolated, to the best of our knowledge, is not hitherto recorded as a host of *C. mycophilum* [22].

***Hypomyces rosellus*** (Alb. and Schwein.) Tul. and C. Tul., Annales des Sciences Naturelles Botanique 13: 12 (1860)

*Anamorph: **Cladobotryum dendroides*** (Bull.) W. Gams and Hooz., Persoonia 6 (1): 103 (1970)

*Strains examined:* Greece: *Attiki:* Athens “National Gardens”, on basidiome of *Flammulina velutipes*, 2008, coll./isol. *Z. Gonou-Zagou*, ATHUM 6847; Mt. Parnitha, in *Abies cephalonica* forest, on basidiome of *Tricholoma* sp., 2008, coll./isol. *Z. Gonou-Zagou* ATHUM 6849; Mt. Parnitha, in *Abies cephalonica* forest, on basidiome of *Hohenbuehelia* sp., 2000, coll./isol. *Z. Gonou-Zagou* ATHUM 7998. *Eurytania:* Ag. Nikolaos, in *Platanus orientalis* forest with *Quercus* sp., *Abies cephalonica* and *Castanea sativa* (sporadically), on ascocarp of *Helvella lacunosa*, 2010, coll./isol. *Z. Gonou-Zagou*, ATHUM 7999. *Karditsa:* Ag. Nikolaos, Mt. Zigourolivado, in *Fagus sylvatica* forest, on polypore basidiome, 2009, coll. *P. Delivorias*, isol. *A. Liakouri* ATHUM 6909. *Xanthi:* Mt. Leivaditis, in *Fagus sylvatica* forest with *Juniperus communis*, on basidiome of *Polyporus varius*, 2009, coll. *A. Sergentani*, isol. *Z. Gonou-Zagou*, ATHUM 6848.

*Mycelium:* felt-like to floccose, somewhat powdery, or with somewhat radially arranged floccose-powdery tufts; whitish, buff, progressing towards ochre, pinkish hues, and various red hues with purplish and brownish undertones; symmetrically spreading, with regular margin (Figure 6a). *Colony reverse:* buff to ochre, progressing towards ochre and various red hues with brownish and purplish undertones (Figure 6b). *Conidia:* ovoid, broadly ellipsoidal to mainly ellipsoidal or cylindrical; (16)21.5–28(35) × (8.5)10–11.5(13) μm, Qm 2.3; two-, three- or four-celled, with prominent apiculus; hyaline, smooth; few slightly curved; mature conidia sometimes constricted at septa; some joined in twos at base or forming short bendy chains (Figure 6h–k). *Sclerotioid structures*: not observed.

*Comments:* Progressive pigmentation of both the colony and substrate from ochre into red hues is one of the main macromorphological traits of the pigment (aurofusarin)-producing *C. dendroides*. Sympodial conidiogenesis is the main diagnostic micromorphological character of all strains of *C. dendroides* (Figure 6d–g), setting it apart from other *Cladobotryum* species. In addition, the long, three- to four- celled conidia contribute to the distinction of the species. The hosts of the Greek specimens of *C. dendroides*, belonging to the basidiomycetous species, *Flammulina velutipes* and *Hohenbuehelia* sp., as well as to the ascomycetous *Helvella lacunosa*, are not recorded so far as hosts of *C. dendroides*, as far as our knowledge goes [22].

***Cladobotryum rubrobrunnescens*** Helfer, Libri Botanici 1: 55 (1991)

*Specimen examined:* Germany, on *Inocybe* sp., 1989, *A. Resinger*, CBS 176.92, ex-type strain.

*Mycelium:* felt-like to compact cottony; whitish, yellow, buff, progressing towards ochre and various red hues with pinkish and purplish undertones; when mature copper-coloured, either in scattered patches or at margin; symmetrically spreading, with regular margin (Figure 7a). *Colony reverse:* buff, progressing towards ochre and various red hues with pinkish, purplish, or brownish undertones (Figure 7b). *Conidia:* ellipsoidal or fusiform to cylindrical to bacilliform, rarely constricted at septa; in slide culture (21)22–26 (29) × (3.5)4–5 (5.5) μm, Qm 5.4, in petri dish culture (14)17–26(29) × (5.5)6–8.5(9.5) μm, Qm 2.9; mainly two-celled, sometimes the basal cell swollen, also one-celled or rarely three- or four-celled; hyaline, smooth; sometimes mature conidia joined in twos (Figure 7e–j). *Sclerotioid structures:* not observed.

***Cladobotryum tenue*** Helfer, Libri Botanici 1: 57 (1991)

*Specimen examined:* Germany: *Regesburg-Keilberg*, on agaric., 1986, *H. Besl*, CBS 152.92, ex-type strain.

*Mycelium:* floccose to sparsely cottony; whitish, yellow, buff, progressing towards pale ochre to pinkish, rarely to various red hues; symmetrically spreading, with regular margin (Figure 8a). *Colony reverse:* buff, progressively turning from ochre to bright terracotta and various red hues with pinkish and purplish to brownish undertones (Figure 8b). *Conidia:* ellipsoidal to clavate, cylindrical to bacilliform, rarely constricted at septa; in slide culture (21)22–28.5(36) × (3.5)4.5–5.5(6) μm, Qm 5.5, in petri dish culture 23–34.5(38) × 4–6, Qm 6.4; mainly two-celled, rarely one- or three-celled, hyaline, smooth (Figure 8f–l). *Sclerotioid structures*: present (Figure 8m).

*Comments:* There are only the protologue descriptions of *C. rubrobrunnescens* and *C. tenue* ever made so far [7]. In terms of macro- and micromorphology, our observations on PDA culture generally comply with the original descriptions, where the medium malt peptone agar (MPA) was used. Some fluctuations in spore shapes and sizes can be attributed to the species variability due to the different culture medium and conditions used. In slide culture of *C. rubrobrunnescens*, the conidia are distinctively much narrower. As the main diagnostic microscopic features of the two species in petri dish culture, the longer and narrower conidia can be considered, as well as the simpler conidiophores of *C. tenue*.


***Cladobotryum* sp.**


Strain examined: Greece: *Fthiotida:* Mesochori, Mt. Oiti, in *Quercus frainetto*, *Q. pubescens* and *Q. coccifera* forest with *Abies cephalonica* (sporadically), on basidiome of *Laccaria laccata*, 2008, coll./isol. *Z. Gonou-Zagou* ATHUM 6851.

*Mycelium:* uneven in terms of texture and pigmentation, patchy-like with cottony and compactly felt-like areas, irregularly distributed; ranging from white-yellowish to buff, pale ochre, pink, progressing towards dark red with pinkish undertones; asymmetrically spreading, with irregular margin (Figure 9a). *KOH reaction:* positive. *Growth rate:* relatively low. *Colony reverse:* buff, progressively turning from ochre to various red hues with purplish undertones (Figure 9b). *Conidiophores:* very long, delicate, subverticillate, unbranched or sparsely branched, unilaterally or dichotomously at conidiophore apices, sometimes emerging almost orthogonally to conidiophore axis; septate; hyaline (Figure 9c–e). *Conidiogenous cells:* elongated, delicate, narrow-cylindrical to almost filiform, subulate, with obtuse apices; arranged in verticils at conidiophore apices, 1–4 per verticil; aseptate, rarely septate; hyaline (Figure 9c–e). *Conidiogenesis:* blastic; a single to multiple conidiogenous loci per conidiogenous cell and a single conidium or up to 5(6) conidia, mainly 3, in a wreath-shaped arrangement (Figure 9c–e). *Conidia:* in petri dish culture ellipsoidal to cylindrical or fusiform to rarely bacilliform; (17.5)19–28(32) × (5.5)6.5–8(10) μm, Qm 3.2; mainly two-celled, rarely three- or four-celled; in slide culture narrowly ellipsoidal to cylindrical or bacilliform; (23)26–33(41) × (4.5)6–7(8.5) μm, Qm 4.8; mainly two-celled, rarely three-celled; hyaline, smooth; basal apiculi central or slightly eccentric (Figure 9f–i). *Sclerotioid structures*: rarely present, mainly as short chains of globose to ellipsoidal cells, thick-walled.

***Cladobotryum*** sp.

Strain examined: Greece: *Fthiotida:* Mesochori, Mt. Oiti, in *Quercus frainetto*, *Q. pubescens* and *Q. coccifera* forest with *Abies cephalonica* (sporadically), on basidiome of *Lactarius salmonicolor*, 2008, coll./isol. *Z. Gonou-Zagou*, ATHUM 6852.

*Mycelium:* cottony to compactly felt-like; white at first, then buff, progressing towards pinkish hues; sporadically white aerial loose tufts; symmetrically spreading, with irregular margin (Figure 10a). *Growth rate:* moderate. *Colony reverse:* buff, progressively turning from terracotta to various red hues with brownish and purplish undertones (Figure 10b). *Conidiophores:* very long, unbranched to sparsely branched in a unilateral manner at conidiophore apices (Figure 10c–f). *Conidiogenous cells:* elongated, narrow-cylindrical, subulate, with obtuse apices; arranged in verticils 1–4(5) per verticil, mainly 3–4 at the apex; aseptate, rarely septate; hyaline (Figure 10c–f). *Conidiogenesis:* blastic; a single to multiple conidiogenous loci per conidiogenous cell and a single conidium or up to 6 conidia, mainly 3, in a wreath-shaped arrangement (Figure 10c–f). *Conidia:* in petri dish culture ovoid, mainly ellipsoidal to cylindrical; (15.5)19–26(28) × 6–10 μm Qm 2.8; sometimes slightly swollen at base and constricted in septal area; mainly two-celled, also three-celled, but rarely four-celled; hyaline, smooth; many germinated; quite often in maturity disintegration of one cell (out of the two) or separation of the two cells; many variably connected in twos or threes at their base or side, rarely forming short chains; possible secondary production of conidia from germinated cell of conidium; in slide culture fusiform to narrowly ellipsoidal to cylindrical; (14)17–23(30) × 5–8 μm, Qm 3.2; mainly two-celled. (Figure 10f–k). *Sclerotioid structures*: present, mainly as chains of globose to ellipsoidal cells, thick-walled.


***Cladobotryum* sp.**


Strain examined: Greece: *Fthiotida:* Mesochori, Mt. Oiti, in *Quercus frainetto*, *Q. pubescens* and *Q. coccifera* forest with *Abies cephalonica* (sporadically), on basidiome of *Russula* sp., 2008, coll./isol. *Z. Gonou-Zagou*, ATHUM 6853.

*Mycelium*: cottony to loosely felt-like; whitish, buff, progressing towards pale ochre to pinkish hues; when mature, secondary whitish cottony at centre (Figure 11a). *KOH reaction:* positive. *Growth rate*: relatively high. *Colony reverse*: buff, progressively turning from ochre to orange and red hues with pinkish undertones (Figure 11b). *Conidiophores:* very long, delicate, unbranched or sparsely branched in a unilateral manner at conidiophore apices or solitary, shorter, emerging orthogonally from very long hyphae; septate; hyaline (Figure 11c–e). *Conidiogenous cells*: elongated, delicate, narrow-cylindrical to almost filiform, subulate, with obtuse apices; arranged in verticils, 1–4(5) per verticil, mainly 3–4 at the apex, sometimes emerging orthogonally to conidiophore axis or in a dichotomous manner at the apex; aseptate, sometimes septate; hyaline (Figure 11c–e). *Conidiogenesis:* blastic; a single to multiple conidiogenous loci per conidiogenous cell and a single conidium or up to 3 conidia in a wreath-shaped arrangement (Figure 11c–e). *Conidia*: in petri dish culture ellipsoidal to cylindrical or fusiform; (15)18–29(32) × (5.5)6.5–9(10.5) μm, Qm 2.9; mainly two-celled, sometimes restricted at septum or basal cell swollen, also three-celled, very rarely four-celled; sparsely curved; quite often joined in twos or threes at their base or laterally; in slide culture narrowly ellipsoidal to cylindrical or rarely bacilliform; (15.5)19.5–26.5(32) × (5)5.5–6.5(7) μm, Qm 3.9; mainly two-celled, very rarely three- to four-celled; hyaline, smooth (Figure 11f–h). *Sclerotioid structures*: present, cells globose to elongated, slightly thick-walled, forming monilioid chains, usually branched (Figure 11i).


***Cladobotryum* sp.**


Strain examined: Greece: *Karditsa:* Ag. Nikolaos, Mt. Zigourolivado, in *Fagus sylvatica* forest, on basidiome of *Lactarius* sp., 2009, coll. *P. Delivorias*, isol. *A. Liakouri* ATHUM 6904. 

*Mycelium*: compactly felt-like to cottony, with white tufts; buff, progressing towards pale ochre and pinkish hues; asymmetrically spreading, with irregular margin (Figure 12a). *KOH reaction:* negative or faintly positive. *Growth rate:* slow to moderate. *Colony reverse*: buff, progressively turning from ochre to red hues with purplish undertones (Figure 12b). *Conidiophores:* very long, delicate, unbranched or very sparsely branched in a unilateral manner at conidiophore apices or solitary, shorter, emerging from very long hyphae; septate; hyaline (Figure 12c–e). *Conidiogenous cells*: elongated, sometimes very elongated, delicate, narrow-cylindrical to almost filiform, subulate; arranged in verticils, 1–3(4) per verticil, mainly 2–3 at the apex, sometimes emerging orthogonally to conidiophore axis or in a dichotomous manner at the apex; aseptate; hyaline; (Figure 12c–e). *Conidiogenesis:* blastic; a single to multiple conidiogenous loci per conidiogenous cell and a single conidium or up to 2 conidia in a V-shaped arrangement (Figure 12c–e). *Conidia*: in petri dish culture ellipsoidal to cylindrical; (14.5)17.5–25(31) × (2.5)4.5–7.5(8.5) μm, Qm 3.7; mainly two-celled, rarely one-celled; basal cell often swollen; sometimes restricted at septum; in slide culture narrowly ellipsoidal to cylindrical or bacilliform; (16.5)19.5–26(32) × (3.5)4.5–5.5(6) μm, Qm 4.7, mainly two-celled, rarely one- to four-celled; hyaline, smooth; (Figure 12f–j). *Sclerotioid structures*: not observed.


***Cladobotryum* sp.**


Strain examined: Greece: *Attiki:* Mt. Parnitha, in *Abies cephalonica* forest, on basidiome of *Tricholoma atrosquamosum*, 2009, *L. Tsampiras*, *A. Liakouri* ATHUM 6912.

*Mycelium*: felt-like to compactly cottony; whitish, buff, progressing towards pale ochre to brownish with pinkish hues; symmetrically spreading, with regular margin (Figure 13a). *KOH reaction:* negative or faintly positive. *Growth rate:* relatively slow to moderate. *Colony reverse*: buff, progressively turning from ochre to bright and dark orange-red hues (Figure 13b). *Conidiophores:* very long, unbranched, sometimes shorter, emerging orthogonally from very long hyphae; septate; hyaline (Figure 13c,d). *Conidiogenous cells*: very elongated, narrow-cylindrical to filiform, subulate, with obtuse apices; in verticils at conidiophores apices, 1–3 per verticil; aseptate, hyaline (Figure 13c,d). *Conidiogenesis:* blastic; a single to multiple conidiogenous loci per conidiogenous cell and a single conidium or up to 3 conidia (Figure 13c,d). *Conidia*: in petri dish culture fusiform to cylindrical or bacilliform, rarely ellipsoidal; (17.5)20–24.5(28) × (4.5)5–6.5(7) μm, Qm 3.8; in slide culture cylindrical to filiform; (21)24–38.5(44.5) × 3.5–5 μm, Qm 7.1; one-celled or two-celled; hyaline, smooth; (Figure 13e–g). *Sclerotioid structures*: not observed.


***Cladobotryum* sp.**


Strain examined: Greece: *Attiki:* Mt. Parnitha, in *Abies cephalonica* forest, on basidiome of *Lactarius salmonicolor*, 2009, *L. Tsampiras*, *A. Liakouri* ATHUM 6913.

*Mycelium*: uneven in terms of texture and pigmentation, with cottony and compactly felt-like areas irregularly distributed, ranging from white to buff, pale ochre, pink; asymmetrically spreading, with irregular margin (Figure 14a). *KOH reaction:* negative. *Growth rate:* relatively slow. *Colony reverse*: buff, progressively turning from ochre to terracotta and various red hues with brownish undertones (Figure 14b). *Conidiophores:* very long, unbranched or sparsely branched in a very irregular and somewhat unilateral or dichotomous manner at conidiophore apices, sometimes shorter, emerging almost orthogonally from very long hyphae; septate; hyaline (Figure 14c). *Conidiogenous cells*: very narrow-cylindrical to almost filiform, subulate, with obtuse apices; arranged in verticils at conidiophores apices, 1–4 per verticil; aseptate; hyaline (Figure 14c). *Conidiogenesis:* blastic; a single or multiple conidiogenous loci per conidiogenous cell and a single conidium or up to 2 conidia in a V-shaped arrangement (Figure 14c). *Conidia*: in petri dish culture ellipsoidal to cylindrical, fusiform to bacilliform; (17.5)24–34.5(43.5) × (4)5–7(10), Qm 4.6; mainly two-celled, but also one-celled; with or without restriction in the septal area; many germinated; in slide culture ellipsoidal to cylindrical, fusiform to bacilliform or filiform; (19.5)22.5–34.5(51.5) × (3)4.5–5.5(6.5) μm, Qm 6.4; mainly two-celled, but also one-celled; hyaline, smooth; (Figure 14d–f). *Sclerotioid structures*: not observed.


***Cladobotryum* sp.**


Strain examined: Greece: *Attiki:* Mt. Parnitha, in *Abies cephalonica* forest, on basidiome of *Cortinarius* sp., 2009, coll. *L. Tsampiras*, isol. *A. Liakouri* ATHUM 6914.

*Mycelium*: uneven in terms of texture and pigmentation, with compactly felt-like and floccose to cottony areas irregularly distributed, ranging from whitish, buff, pale ochre to red hues with pinkish undertones; asymmetrically spreading, with irregular margin (Figure 15a). *KOH reaction:* positive. *Growth rate:* relatively low. *Colony reverse*: buff, progressively turning from ochre to various red hues with pinkish undertones (Figure 15b). *Conidiophores:* very long, unbranched or very sparsely branched in a somewhat unilateral manner at conidiophore apices; septate; hyaline (Figure 15c,d). *Conidiogenous cells*: very narrow-cylindrical to almost filiform, subulate, with obtuse apices; in verticils on conidiophores apices, 1–3 per verticil; aseptate; hyaline (Figure 15c,d). *Conidiogenesis*: blastic; a single or multiple conidiogenous loci per conidiogenous cell and a single conidium or up to 2 conidia in a V-shaped arrangement (Figure 15c,d). *Conidia*: in petri dish culture ovoid, ellipsoidal to cylindrical; (16)17.5–22.5(26.5) × (5)7–8.5(10) μm, Qm 2.7; mainly two-celled, rarely one-celled or three–four-celled; many with restriction in the median septum; some joined in twos, or more, at their base or laterally; in slide culture fusiform to narrowly ellipsoidal or cylindrical; (12.5)15.5–20.5(22.5) × 5–8 μm, Qm 2.8; mainly two-celled (Figure 15e,f). *Sclerotioid structures*: rarely globose to elongated cells, thick walled, forming short chains.

*Comments:* All unidentified strains are characterised by the final reddish hues of both the mycelium and the colony reverse. They were isolated from various agaricoid basidiomes that come from three different geographic locations and habitats. Their macroscopic characters seem quite similar, that is, the colony is more or less felty to cottony, and yellow, buff to pinkish, while their microscopic ones exhibit similarities, and many overlaps as well as differences. They differ clearly from our strains of *C. dendroides* and *C. mycophilum* and exhibit similarities with *C. rubrobrunnescens* and *C. tenue* [7]. The strains, ATHUM 6912, 6913, 6914—coming from the same collecting site—are similar in terms of colony macromorphology, but the strain, ATHUM 6914, is differentiated considering spore shape and size from the other two that have narrower and longer spores. In addition, the strain, ATHUM 6914, shares resemblances in micro- and macromorphology with the strain, ATHUM 6852, which is isolated from a different site. The strains, ATHUM 6851 and ATHUM 6853, as well as ATHUM 6904, can be grouped together considering their micromorphological similarities, although they individually exhibit some deviations. It is worth noting that all the above strains were also cultured in slides, where the spores always appeared narrower and, in most cases, quite longer than the corresponding ones in the petri dish.

Figure 16 provides clear visual information on the grouping of the species according to their conidial morphology when grown in petri dish cultures with PDA. On one extreme, there are conidia approaching broader ellipsoidal shapes, as seen in the species *C. varium*, *C. verticillatum*, *C. mycophilum*, and *C. dendroides*; all of them belong to different size categories, with *C. varium* being distinctly smaller, while the two latter being clearly larger and resembling one another. On the other extreme, there are conidia exhibiting narrower ellipsoidal shape, as seen in the unidentified strain ATHUM 6913, and especially in the ex-type species *C. tenue*. In between are all the other species/strains, with *C. fungicola* being closer to a broader ellipsoidal shape and noticeably smaller, therefore clearly apart from the rest of the same ‘shape category’. Conidia of the strains, ATHUM 6904 and ATHUM 6912, are somewhere in-between the two extremes—considering both their shape and length—in comparison to the rest of the strains that are grouped together as a ‘cluster’. 

### 3.2. Molecular Analyses

A phylogenetic approach based on the ITS region (ITS1-5.8S-ITS2) was implemented to clarify the relationships among the 29 examined Greek strains and the 2 ex-type strains (*C. rubrobrunnescens* and *C. tenue*) (Figure 17). In order to resolve any phylogenetic ambiguity, 63 publicly available sequences were also included, representing all known ITS sequences of the *Cladobotryum* species to date. Discrimination between the species is observed with high support (Posterior Probability value from 80 to 100%). Although the strains are well grouped at the species level, there is great intraspecific variability in most cases. As expected, Greek strains are distributed across the phylogenetic tree. In detail, *C. fungicola* is grouped with *H. semitranslucens*, which is the ex-type containing both states of reproduction (asexual/anamorphic and sexual/teleomorphic) with 100% support since their ITS sequences are identical. These species, with the currently available data, diverged in earlier evolutionary stages [84]. *Cladobotryum varium* and its associated teleomorph *H. aurantius* strains are grouped as a monophyletic clade next to *C. heterosporum*, for which a red pigment production is referred to in the available literature [36]. If the pigments, rugulosin, skyrin, and emodin, for the currently examined species of the genus *Cladobotruym* are considered, then the metabolite rugulosin seems to be a feature of the species, *C. varium* and *C. fungicola*, while skyrin and emodin can be found mainly in *C. varium* and probably in the cluster of *C. heterosporum* and *C. amazoense* [7,85]. Even though species *C. apiculatum* and *C. verticillatum* are sister clades with 87% PP support value, metabolite norbikaverin is produced only in *C. verticillatum*, and this latter species is grouped with the respective teleomorph species *H. armeniacus* with high support (99% PP value). Pigment bikaverin is produced in *C. varium* as described earlier [33]. The remaining species exhibiting red pigmentation, presumably due to aurofusarin, diverged later [7,36,59,60,85]. *Cladobotryum mycophilum* along with *H. odoratus* form a monophyletic clade that is basal to the rest of the examined species producing aurofusarin. The metabolites, rosellisin and rosellisin-aldehyde, are particularly produced by *C. dendroides* strains, which are grouped as two subclusters that are basal to the *C. tenue* and *C. rubrobrunnescens* clade, and also in *C. varium* [86,87]. Greek *Cladobotryum* spp. strains (URPs) are clustered along with *C. tenue* and present a similarity with *C. rubrobrunnescens* as a basal clade, but their relationship is unclear since the support values (PP) of these topologies are non-significant. However, ITS-based phylogeny shows the cluster of a large group that contains *C. tenue*, *C. rubrobrunnescens*, and the Greek *Cladobotryum* spp. strains (Figure 17).

### 3.3. NMR-Based Metabolomics/Chemotaxonomy

In order to employ a more integrative approach to the taxonomic investigation of Greek strains of the genus *Cladobotryum*, in conjunction with morphological and molecular analyses, an attempt to utilise ^1^H NMR-based metabolomics as a chemotaxonomic tool was made. Spectroscopic data were transformed to arithmetic values and subjected to multivariate statistical analysis.

The score plot of the unsupervised principal component analysis (PCA) is presented in Figure 18a. The generated model with 15 components and with R2X(cum) = 0.78 and Q2(cum) = 0.54 satisfactorily described the data variation. Three clusters were observed by visual inspection of the generated model, and, more specifically, Cluster I is comprised of the strains corresponding to the species *C. apiculatum*, *C. verticillatum*, and a small number of *C. mycophilum* strains; Cluster II encompasses the strains corresponding to the species *C. varium* and *C. fungicola;* while Cluster III involves the rest of the *C. mycophilum* strains, as well as those corresponding to the species *C. dendroides*, *C. rubrobrunnescens*, *C. tenue*, and the URPs. Cluster I is less populated and tight. Clusters II and III are more populated and the distribution of their samples along the PC1 axis suggests a higher variance of their spectral characteristics. Nevertheless, Clusters II and III are clearly separated along the PC2 axis.

Further analysis was performed using the species information as a Y variable with the aim to reveal species-specific spectral characteristics responsible for the clustering. The score plot of Projection to Latent Structures (PLS-DA) is shown in Figure 18b. The created model had 7 components with R2X(cum) = 0.57 and Q2(cum) = 0.48, while the R2Y(cum) = 0.68. The model was validated through a permutation test (100 permutations; Appendix A). The supervised analysis based on species corroborates the observations of the PCA, suggesting an existing intraspecies chemical similarity, thereby further separating *C. fungicola* strains from those corresponding to *C. varium* (Figure 18b). It is interesting to note that on the second axis, macroscopically-observed pigmentation differences of the species/isolates are projected, with the red-pigmented ones being located above and the non-red being located below x-axis.

A more thorough investigation was performed for the red-pigmented species, which aimed at a more detailed description of the spectroscopic profile of the URPs, and tried to possibly unravel the similarities and differences hidden within a wider context. The PCA and PLS-DA score plots are shown in Figure 19a,b, respectively. The model that resulted from PCA had 10 principal components and described 76% of the variability, while predicting a lower share (51%). The score plot suggests the spectral similarity of *C. mycophilum* and *C. dendroides* species. URPs are differentiated from *C. mycophilum* and *C. dendroides* along the 1st axis, indicating the existence of species-specific spectral characteristics. However, in the centre of the T^2^ Hotteling ellipse, samples from all strains are overlapped and only *C. rubrobrunnescens* is situated in the 2nd axis, at the margins of the ellipse. The supervised analysis with 6 components and R2X(cum) = 0.58, R2Y(cum) = 0.87, and Q2(cum) = 0.66 distinguishes *C. mycophilum* from *C. dendroides* species in the 2nd component, while in the 1st component of the model, the difference of URPs with the rest of the samples is projected (Figure 19b). *Cladobotryum rubrobrunnescens* and *C. tenue* ex-type strains are located at two edges of *C. dendroides* strains, exhibiting a more distant spectroscopic profile when analysed within the red-pigmented strains. Similarly, with the unsupervised analysis, a sub-group of URPs is found together with *C. tenue* isolates in close vicinity of *C. dendroides* strains. The model was validated using 100 random permutations (Appendix A). The Variable Importance in Projection scores (VIPs) of the 1st component—the axis separating URPs from the other examined red-pigmented species—reveals sugar moieties as the major characteristic of URPs.

Τhe ^1^H NMR spectra of the hydroalcoholic extracts of the isolates are presented in Figure 20 and grouped according to the results obtained from the PCA analysis (Figure 18a) to facilitate visual inspection. More precisely, panel a shows the ^1^H NMR spectra of the isolates corresponding to the species *C. mycophilum*, *C. verticillatum*, *C. apiculatum*, and *C. fungicola*; panel b shows the isolates of the species *C. varium*; and panel c shows the isolates of the species *C. dendroides*, *C. tenue*, *C. rubrobrunnescens*, and the URPs. The investigated isolates exhibited rich metabolic fingerprints with dominant resonances characteristic of sugar moieties (6.00–3.00 ppm, with anomeric protons observed at 5.50–4.50 ppm as doublets), as well as polyols (3.90–3.60 ppm), among which mannitol appeared to be the prevailing one. Glucose, both in α and β configuration, was observed in a relatively high concentration in all *C. varium* strains when compared to other isolates. In *C. dendroides*, and especially in URPs, glucose also had a prominent presence. Acetic acid appeared as a prominent singlet at 1.89 ppm with a high inter- and intraspecies variability. In the majority of the spectra, an additional characteristic singlet at 2.15 ppm is indicative of the presence of *N*-acetylated compounds. Additional singlets related to choline derivatives (3.25–3.20 ppm) were detected close to the region where deuterated methanol and its satellites resonate. However, a relatively low concentration of these molecules allowed only the observation of *N*-trimethyl resonances—which are peaks of low diagnostic value—compared to the rest of the protons of choline derivatives, which exhibited characteristic complex patterns that, otherwise, would have facilitated peak annotation. The presence of a strong, featureless peak around 1.20 ppm, in combination with a triplet in the region of 0.95–0.85 ppm, provided evidence for the presence of long fatty acid moieties (methylene groups at 1.20 ppm and end-methyls at 0.80 ppm). These features are observed in several samples (one *C. mycophilum* strain, *C. fungicola*, two *C. varium* strains, few *C. dendroides*, and URPs). Additionally, in the same spectral region, the characteristic methyl doublet of alanine at 1.48 ppm and lactic acid at 1.31 ppm were also observed. Finally, among *Cladobotryum* spp., a wide distribution of a *trans* double bond region (7.80–6.60 ppm, *J_HH_* ~ 15 Hz) was observed, which is potentially explained by the presence of a characteristic group of fungal secondary metabolites that are structurally related to rosellisin. Lastly, out of six isolates of *C. dendroides*, two were able to produce a significant amount of phomalactone, characterised by a detailed analysis of 1D and 2D NMR spectra (Appendix A).

## 4. Discussion

### 4.1. Contribution of Data Derived from Morphology and Notes on Ecology

The studied strains of the species *C. fungicola*, *C. apiculatum*, *C. verticillatum*, *C. varium*, *C. mycophilum* and *H. rosellus*/*C. dendroides*—due to their distinguishable morphological characters in culture, as well as their ecological features—can be identified with certainty. The colony texture, its obverse and reverse colours, and the morphology of the reproductive structures, either asexual (conidiophores, conidiogenous cells and conidia) or sexual (perithecia and ascospores)—when available—are diagnostic. Some morphological differences could be attributed to the different growth media and conditions used for culturing, since all our strains were grown on PDA, in contrast to a variety of media used in the already referred descriptions of *Cladobotryum* species. As observed from our study, conidia of a strain can vary in shape and size when studied from natural substrates or different growth media or culturing methodologies (slide, solid, and liquid cultures). Even though these deviations can be ascribed to the morphological variability within the genus due to environmental factors, in some cases they may amplify the ambiguity in the species discrimination and taxonomy. It is important when comparing character states to keep constant the factors that can affect the fungal development.

In contrast, the morphological characters of the ‘URP’ *Cladobotryum* strains cannot contribute to a valid species identification. They exhibit a great amount of similarity, with many micromorphological features overlapping and bearing resemblance to the species *C. rubrobrunnescens* and *C. tenue*, a trait verified by the ITS-based phylogenetic analysis. To clarify the identity of these strains, a more thorough study comprising a multi-loci molecular phylogenetic analysis is needed.

A very interesting feature observed in many strains is the formation of variously interconnected conidial chains. In some cases, this phenomenon featured the production of a secondary conidium, emerging directly from a mature primary conidium. In other cases, conidial anastomosis was a probable mechanism behind the chain formation—a claim supported by the observed structures, such as anastomosis-related short tubes and bridges. The phenomenon of chains of fused conidia was first depicted in a drawing of *H. rosellus* by Tulasne and Tulasne [91]. It was also observed in a strain of *C. varium* by Arnold and Yurchenko [82]. Roca et al. [92,93,94] described the process that takes place during the fusion of fungal spores and involves the formation and interaction of specialised hyphae, called conidial anastomosis tubes (CATs). CATs can be formed directly from the conidia; they might be induced by conidial density or starvation and reveal gene exchange, thus mediating heterokaryosis or recombination when sexual reproduction is rare or missing. 

As mentioned previously, it seems that there is a taxonomic and nomenclatural issue regarding the combination of anamorphs and their teleomorphs that, in most cases, is resolved by isolations containing both states, either ex-types or other strains [6,7,8,9,10,11,12]. Molecular analyses of such isolations compared to those of solely anamorphic or teleomorphic strains can corroborate the interconnections. Interestingly, our strain of *C. mycophilum* ATHUM 8001 could be useful for such a comparison since it represents a holomorph containing both the teleomorph and anamorph. Therefore, the establishment of the current names is needed according to the ICN, following the “one fungus, one name” principle. In the case of most *Cladobotryum/Hypomyces* species listed in the mycological online databases MycoBank (https://www.mycobank.org, accessed on 28 July 2022) and Index Fungorum (http://www.indexfungorum.org, accessed on 28 July 2022), no such combination is mentioned, and the anamorph and teleomorph names are treated separately, with the only exception being *C. dendroides/H. rosellus.*

All identified species, besides *C. apiculatum* and *C. verticillatum*, do not exhibit a host preference, inhabiting mainly agaricoid basidiomes, whereas *C. apiculatum* and *C. verticillatum* are characterised by their preference for russuloid basidiomycetes.

### 4.2. Contribution of Data Derived from Molecular Phylogenetic Analyses

The phylogenetic analysis shows that the examined strains isolated from Greek habitats are spread across the phylogenetic tree among strains that originated from tropical or other climates. Both sequence and metabolic data suggest that the cluster including the species, *C. fungicola*, diverged early phylogenetically and presents an “ancestral”-like status (Figure 17). The appearance of red pigmentation, presumably due to aurofusarin and related pigment production [7], possibly occurred as one major evolutionary event at the divergence of *C. apiculatum* against the earlier diverging species (Figure 17). However, the exclusion of the possible reappearance of these metabolites in many different evolutionary events cannot be overruled. Additionally, the metabolite bikaverin occurs in the *C. varium* species and the *C. verticillatum* species [7,36]. URPs strains belonging to the *C. tenue/rubrobrunnescens* clade do not provide any morphological or phylogenetic information, which could contribute to the species identification as either of the ex-type strain of these two species, and thus, further studies have to be performed in the future for clarifying, beyond any doubt, the taxonomic status of these strains. 

### 4.3. Contribution of Data Derived from NMR-Based Metabolomics

A multivariate analysis of ^1^H NMR spectral data represents a potent chemotaxonomic tool. It offers a holistic representation and a detailed characterization of biological systems by incorporating a large amount of metabolic/metabolome-derived information at a specific time point, regarding both the chemical composition and relative concentration of single components. The importance and the complexity hidden in minor compounds could play a decisive role in the identification process regarding fungi in comparison to the approaches focused on monitoring specific classes of metabolites [95]. The unsupervised principal component analysis (PCA) is indeed used in complex systems to provide an insight into the existence of clustering tendencies and the detection of outliers. On the other hand, supervised discriminant analyses (PLS-DA) and orthogonal PLS-DA (o-PLS-DA), are based on given clustering information and generate models to explain class separability.

Despite being studied for their potentially bioactive compounds, the species of the genus *Cladobotryum* are—to the best of our knowledge—unexplored in the context of large-scale metabolic profiling, regardless of the analytical technique. The results obtained from the chemotaxonomic analysis in the present study—conducted on the basis of ^1^H NMR data—were in line with those acquired via molecular and morphological approaches. The idea of the natural classification of species via application of polyphasic approaches—including those that incorporate multivariate analysis—has been successfully applied over the years in various fungi [51,53,54,96,97].

During the fingerprinting analysis of ^1^H NMR data, two major doublets at 4.47 ppm and 5.10 ppm have been identified and assigned to the pyranose forms of β- and α-glucose units, respectively. Sugar moieties (Figure 20) are related to the cultivation medium and their different abundances reflect the ability of each species to take advantage of available nutrients and to adapt to the existing cultivation conditions. A chemotaxonomic diagnostic character of a metabolite is reflected in its wide but not universal distribution, and a high degree of consistency regarding a variety of exogenous factors, such as carbon source (i.e., growth medium) [48]. It is worth noting that previous findings [50] argue that primary metabolites (PMs) can be as successfully utilised in fungal chemotaxonomy as secondary metabolites (SMs) have been—a context within which they have often been overlooked and characterised as compounds of lower differentiation power compared to SMs, thus being deemed unsuitable for taxonomic purposes [47,48,49]. It is, however, recognised that some of the primary metabolites (e.g., polyols, carbohydrates, lipids)—being a response to ecophysiological environmental factors—exhibit a certain potential to be species-specific [47]. Indeed, during our investigation using ^1^H NMR-based metabolomics, we found a clear separation of tested strains of the genus *Cladobotryum* with respect to their primary metabolites. It is well-known that optimal fungal growth—and hence the assimilation of nutrients and the production of both primary and secondary metabolites—is strongly influenced by the medium used and growth conditions [98,99]. There is an intrinsic interconnection and interdependence between fungal primary and secondary metabolism, since pathways for SMs are directly influenced by those of PMs [98,100]. In addition, the production of certain secondary metabolites is context-dependent or only occurs during a particular life stage of a fungus or under the influence of a particular stimulus, resulting in some SM pathways remaining silent [98]. Therefore, in cases of limited production of SMs, our findings show that—when it comes to the genus *Cladobotryum*—primary metabolites could serve as a valuable chemotaxonomic tool/marker.

Furthermore, the metabolomic analysis offered the possibility to obtain valuable chemical information related to URPs since the ITS-based and morphological approaches—although recognizing the distinguishing features of the URPs—did not make any further differentiations within this group. In fact, URPs, which were represented by the same cluster even in the PCA analysis, showed similarities in the *trans* double bond region of the ^1^H NMR spectra, as well as in the content of primary metabolites (Figure 20c). The distribution of the *trans* double bond region was also observed in several other fungal isolates (*C. rubrobrunnescens* and *C. tenue*) belonging to the same cluster, Cluster III of the PCA (Figure 20c). In its most basic sense, this fact may describe the tight similarity of URPs in terms of certain genetic elements with the species being successfully characterized by molecular and morphological evidence (i.e., *C. tenue*, *C. rubrobrunnescens*, *C. dendroides*). The wide distribution of the *trans* double bond region could be explained by the presence of a characteristic group of secondary metabolites that are structurally related to rosellisin, an acetogenin α-pyrone, which was previously isolated from *H. rosellus*, the teleomorph of *Cladobotryum dendroides* [7]. Rosellisin derivatives are a class of fungal-derived SMs containing a *trans* double bond system, which are widely present and isolated from *Cladobotryum* species [7,86,90]. However, the distribution could notably vary among species. For instance, beside the comparable distribution of common *trans* double bond NMR signals at 7.71 ppm and 6.71 ppm between the URPs and the rest of the strains belonging to Cluster III (Figure 20c), a significantly lesser abundance of signals at 7.40 ppm and 6.76 ppm in the URPs—in combination with the prominent presence of doublets at 7.60 ppm and 6.60 ppm—could explain the common clustering of URPs during PCA analysis, as well as their diversification with respect to the rest of *Cladobotryum* spp.

In conclusion, this study clearly showed that all different approaches, i.e., morphological analyses, molecular typing of single locus, but, most importantly, multi-loci comparisons and metabolomics, need to be combined to safely identify and characterise the different *Cladobotryum* species.

## Figures and Tables

**Figure 1 jof-08-00877-f001:**
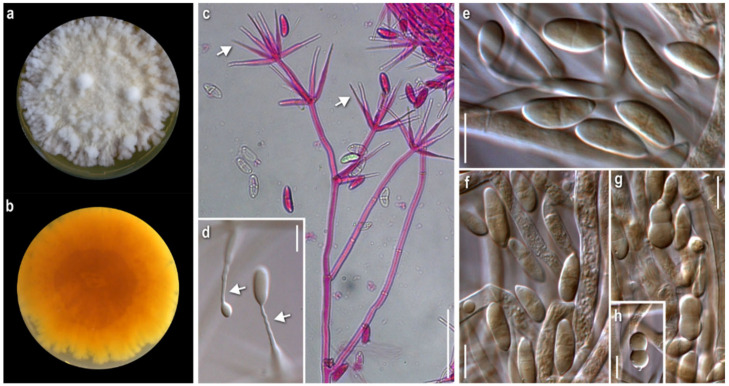
*Cladobotryum fungicola* ATHUM 6855. (**a**) colony obverse; (**b**) colony reverse; (**c**,**d**) conidiophores and conidiogenous cells with the characteristic zig-zag apical part (arrows), (**c**) in phloxine B, (**d**) in Melzer’s; (**e**–**h**) diverse conidia, (**e**,**f**) typical conidia, and (**g**,**h**) conidia with central constriction in Melzer’s. Scale bars: 50 μm = (**c**); 10 μm = (**d**–**h**).

**Figure 2 jof-08-00877-f002:**
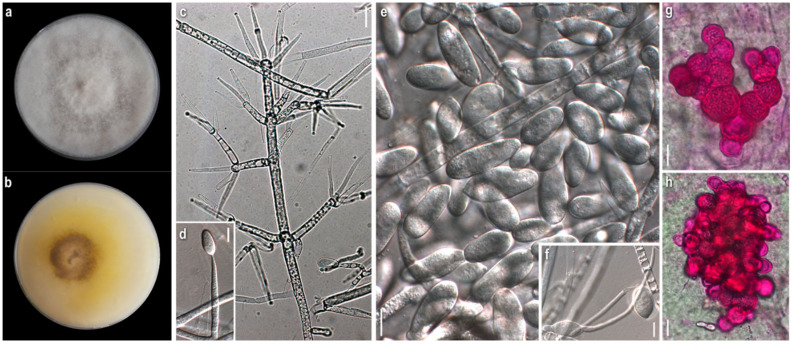
*Cladobotryum apiculatum* ATHUM 6907. (**a**) colony obverse; (**b**) colony reverse; (**c**,**d**) conidiophores and conidiogenous cells in KOH 3%; (**e**,**f**) conidia in KOH 3%; (**e**) typical one-celled (**f**) germinated from both sides with (**g**,**h**) sclerotioid structures in phloxine B. Scale bars: 20 μm = (**c**,**g**,**h**); 10 μm = (**d**–**f**).

**Figure 3 jof-08-00877-f003:**
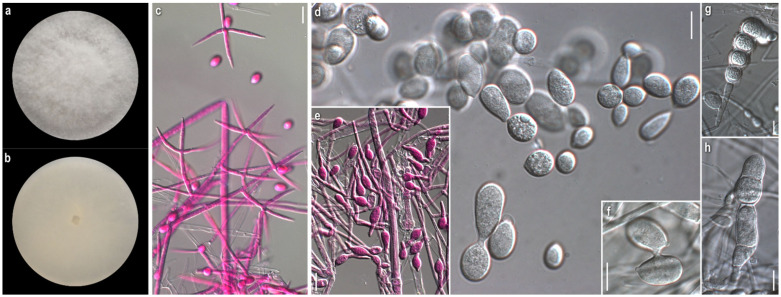
*Cladobotryum verticillatum*. (**a**) colony obverse (ATHUM 6921); (**b**) colony reverse (ATHUM 6921); (**c**) conidiophores and conidiogenous cells (ATHUM 6921) in phloxine B; (**d**–**f**) diverse conidia (ATHUM 6850), (**d**,**f**) conidia joined in twos in KOH 3%, (**e**) germinated conidia in phloxine B; (**g**,**h**) sclerotioid structures with rough (**g**) and smooth (**h**) walls (ATHUM 6850) in KOH 3%. Scale bars: 20 μm = (**c**); 10 μm = (**d**–**h**).

**Figure 4 jof-08-00877-f004:**
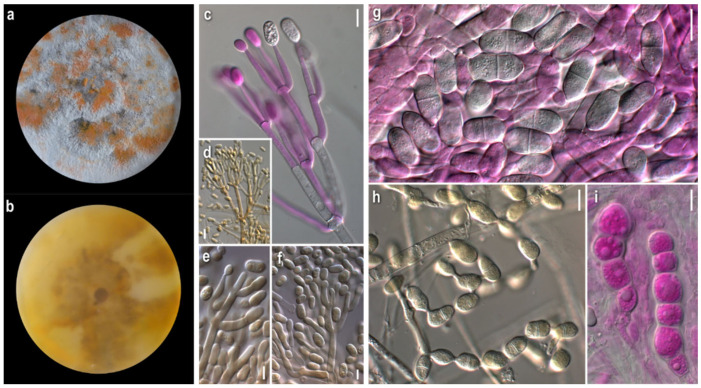
*Cladobotryum varium*. (**a**) colony obverse (ATHUM 8002); (**b**) colony reverse (ATHUM 7995); (**c**–**f**) conidiophores and conidiogenous cells in different stages of maturity, (**c**) immature (ATHUM 6845) in phloxine B, (**d**–**f**) mature (ATHUM 6908) in Melzer’s; (**g**,**h**) conidia, (**g**) typical 2-celled conidia (ATHUM 6845) in phloxine B, (**h**) chains of conidia (ATHUM 6846) in Melzer’s; (**i**) sclerotioid structures (ATHUM 6908) in phloxine B. Scale bars: 20 μm = (**d**); 10 μm = (**c**,**e**–**i**).

**Figure 5 jof-08-00877-f005:**
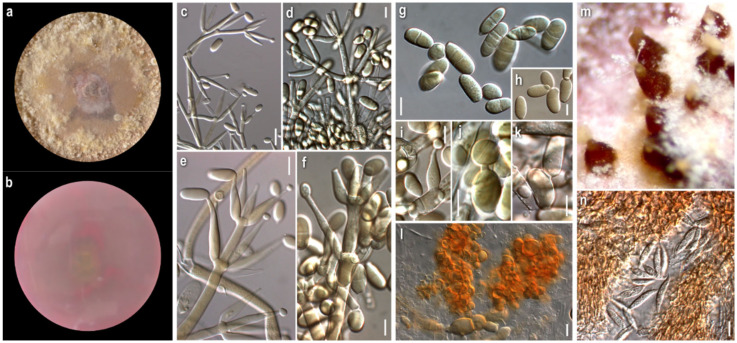
*Cladobotryum mycophilum*. (**a**) colony obverse (ATHUM 8000); (**b**) colony reverse (ATHUM 8000); (**c**–**f**) conidiophores and conidiogenous cells, (**c**,**e**): ATHUM 8000, (**d**): ATHUM 8001, (**f**): ATHUM 6906) in Melzer’s; (**g**–**k**) conidia, (**g**,**h**) typical conidia, (**i**) conidiogenous cell at conidiation, (**j**) conidia joined with short tube, (**k**) two joined conidia from a short chain (**g**,**j**): ATHUM 8001, (**h**), (**i**,**k**): ATHUM 6906) in Melzer’s; (**l**) sclerotioid structures (ATHUM 7994) in Melzer’s; (**m**) perithecia (ATHUM 8001); (**n**) ascospores (ATHUM 8001) in Melzer’s. Scale bars: 20 μm = (**c**,**l**); 10 μm = (**d**–**k**,**n**).

**Figure 6 jof-08-00877-f006:**
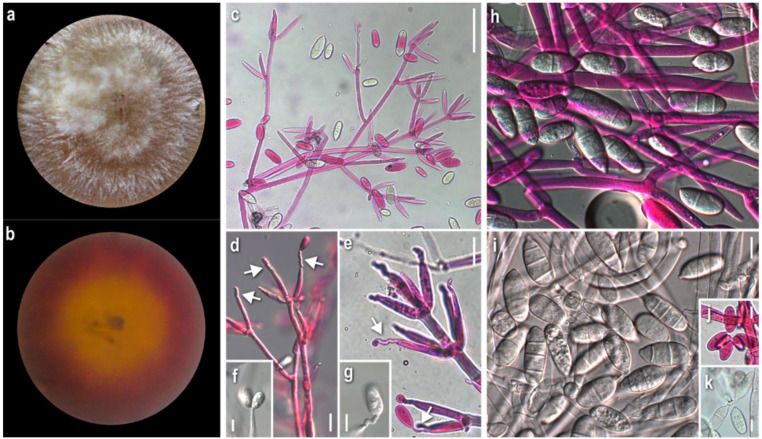
*Cladobotryum dendroides*. (**a**) colony obverse (ATHUM 7999); (**b**) colony reverse (ATHUM 7999); (**c**–**g**) conidiophores and conidiogenous cells, with the characteristic sympodial conidiogenesis (arrows), (**c**,**f**,**g**): ATHUM 6847; (**d**): ATHUM 7996; (**e**): ATHUM 7999), (**c**–**e**) in phloxine B, (**f**,**g**) in KOH 3%; (**h**–**k**) conidia, (**h**,**i**) diverse conidia, (**j**) chain of conidia (**k**) conidia joined in twos, (**h**): ATHUM 6847; (**i**): ATHUM 7999; (**j**), ATHUM 6917; (**k**) ATHUM 6915, (**h**,**j**) in phloxine B, (**i**,**k**) in KOH 3%. Scale bars: 50 μm = (**c**); 20 μm = (**d**,**e**); 10 μm = (**f**–**k**).

**Figure 7 jof-08-00877-f007:**
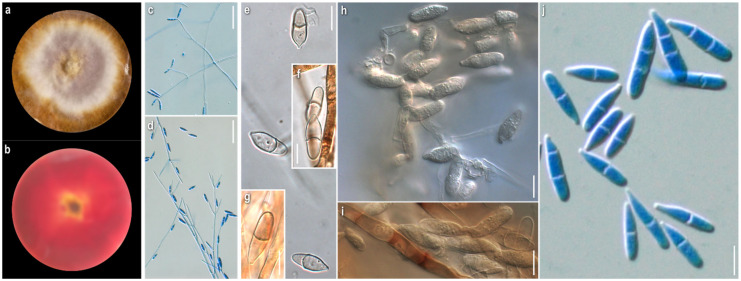
*Cladobotryum rubrobrunnescens* (CBS 176.92). (**a**) colony obverse; (**b**) colony reverse; (**c**,**d**) conidiophores and conidiogenous cells, in lactophenol cotton blue, slide culture; (**e**–**j**) conidia, in Melzer’s, petri dish culture, (**e**–**h**) two- and one-celled, (**i**) joined in twos, (**j**) in lactophenol cotton blue, slide culture. Scale bars: 50 μm = (**c**,**d**); 10 μm = (**e**–**j**).

**Figure 8 jof-08-00877-f008:**
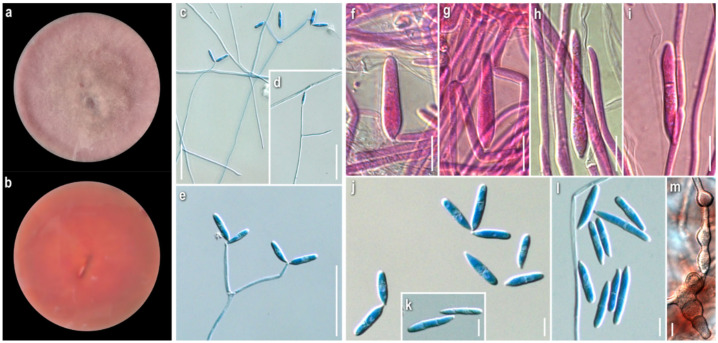
*Cladobotryum tenue* (CBS 152.92). (**a**) colony obverse; (**b**) colony reverse; (**c**–**e**) conidiophores and conidiogenous cells, in lactophenol cotton blue, slide culture; (**f**–**l**) conidia, (**f**–**i**) in phloxine B, petri dish culture, (**j**–**l**) in lactophenol cotton blue, slide culture; (**m**) sclerotioid structures, in Melzer’s, petri dish culture. Scale bars: 50 μm = (**c**–**e**); 10 μm = (**f**–**m**).

**Figure 9 jof-08-00877-f009:**
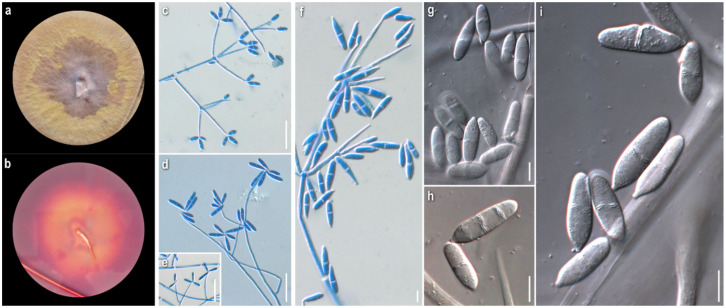
*Cladobotryum* sp. ATHUM 6851. (**a**) colony obverse; (**b**) colony reverse; (**c**–**e**) conidiophores and conidiogenous cells, in lactophenol cotton blue, slide culture; (**f**–**h**) conidia, (**f**) in lactophenol cotton blue, slide culture, (**g**–**i**) in KOH 3%, petri dish culture. Scale bars: 50 μm = (**c**–**e**); 10 μm = (**f**–**i**).

**Figure 10 jof-08-00877-f010:**
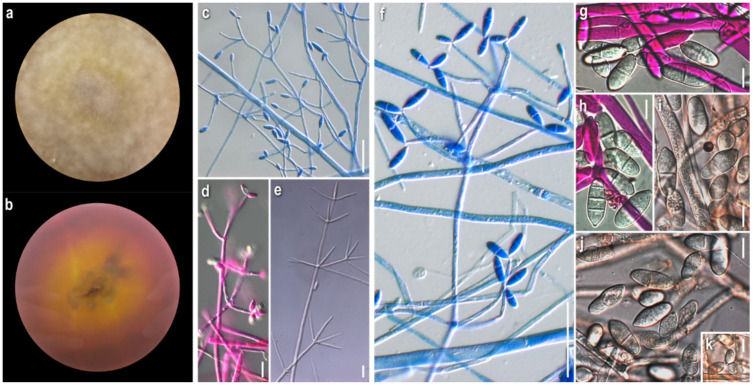
*Cladobotryum* sp. ATHUM 6852. (**a**) colony obverse; (**b**) colony reverse; (**c**–**f**) conidiophores and conidiogenous cells, (**c**,**f**) in lactophenol cotton blue, slide culture, (**d**) in phloxine B, (**e**) in KOH 3%, petri dish culture; (**f**–**k**) conidia, (**g**–**j**) diverse, (**k**) joined in twos (**f**) in lactophenol cotton blue, slide culture, (**g**,**h**) in phloxine B, (**i**–**k**), in Melzer’s petri dish culture. Scale bars: 50 μm = (**c**,**f**); 20 μm = (**d**,**e**); 10 μm = (**g**–**k**).

**Figure 11 jof-08-00877-f011:**
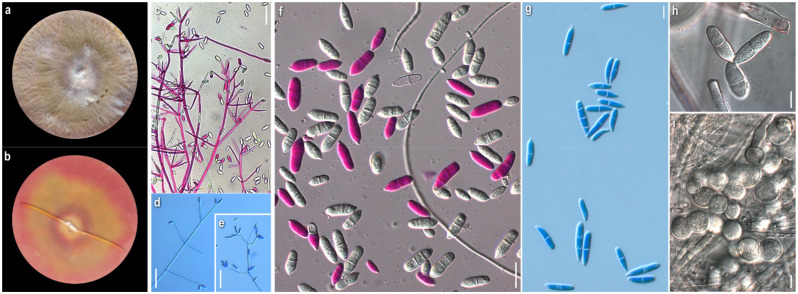
*Cladobotryum* sp. ATHUM 6853. (**a**) colony obverse; (**b**) colony reverse; (**c**–**e**) conidiophores and conidiogenous cells, (**c**) in phloxine B, petri dish culture, (**d**,**e**) in lactophenol cotton blue, slide culture; (**f**–**h**) conidia, (**f**) in phloxine B, (**h**) joined in threes in KOH 3%, petri dish culture, (**g**) in lactophenol cotton blue, slide culture; (**i**) sclerotioid structures, in KOH 3%, petri dish culture. Scale bars: 50 μm = (**c**–**e**); 20 μm = (**f**); 10 μm = (**g**–**i**).

**Figure 12 jof-08-00877-f012:**
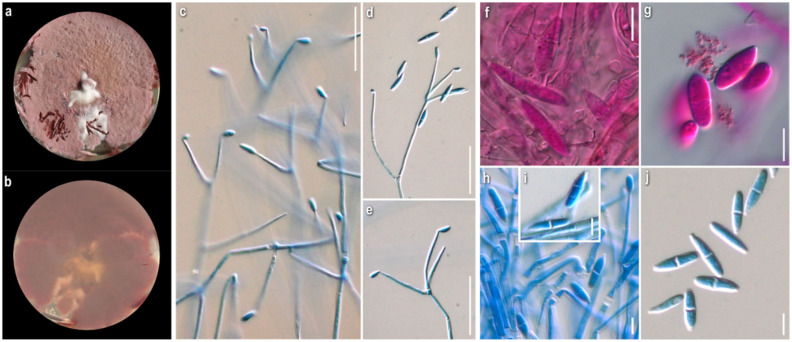
*Cladobotryum* sp. ATHUM 6904. (**a**) colony obverse; (**b**) colony reverse; (**c**–**e**) conidiophores and conidiogenous cells, in lactophenol cotton blue, slide culture; (**f**–**j**) conidia, (**f**,**g**) in phloxine B, petri dish culture, (**h**–**j**) in lactophenol cotton blue, slide culture. Scale bars: 50 μm = (**c**–**e**); 10 μm = (**f**–**i**).

**Figure 13 jof-08-00877-f013:**
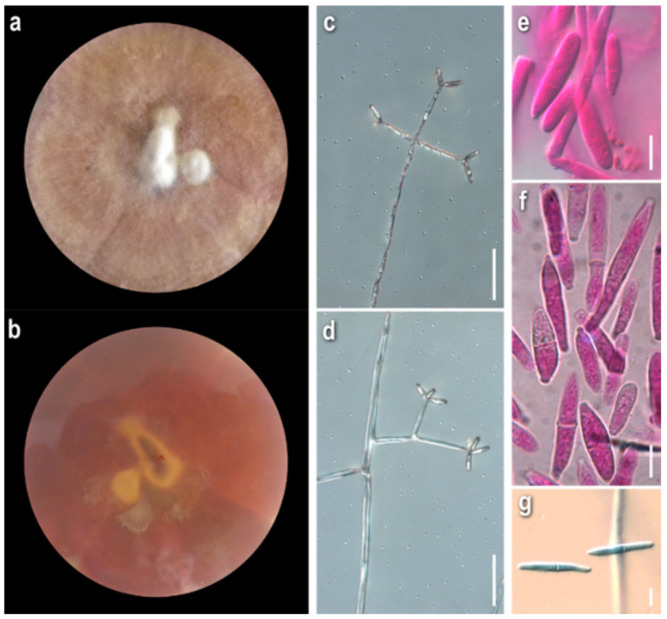
*Cladobotryum* sp. ATHUM 6912. (**a**) colony obverse; (**b**) colony reverse; (**c**,**d**) conidiophores and conidiogenous cells, in lactophenol cotton blue, slide culture; (**e**–**g**) conidia, (**e**,**f**) in phloxine B, petri dish culture, (**g**) in lactophenol cotton blue, slide culture. Scale bars: 50 μm = (**c**,**d**); 10 μm = (**e**–**g**).

**Figure 14 jof-08-00877-f014:**
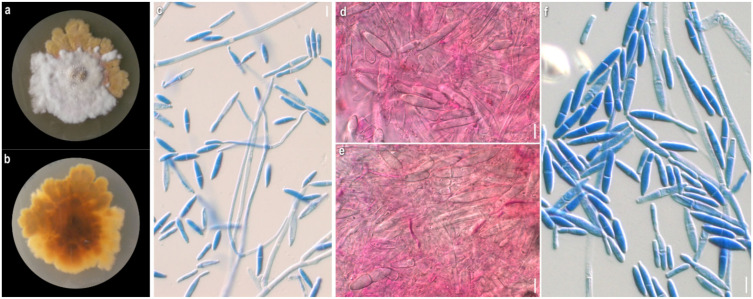
*Cladobotryum* sp. ATHUM 6913. (**a**) colony obverse; (**b**) colony reverse; (**c**) conidiophores and conidiogenous cells, in lactophenol cotton blue, slide culture; (**d**–**f**) conidia, (**d**,**e**) in phloxine B, petri dish culture, (**f**) in lactophenol cotton blue, slide culture. Scale bars: 50 μm = (**c**); 10 μm = (**d**–**f**).

**Figure 15 jof-08-00877-f015:**
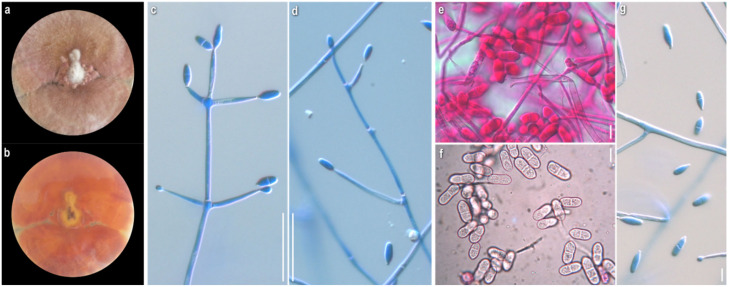
*Cladobotryum* sp. ATHUM 6914. (**a**) colony obverse; (**b**) colony reverse; (**c**,**d**) conidiophores and conidiogenous cells, in lactophenol cotton blue, slide culture; (**e**–**g**) conidia, (**e**) in phloxine B, (**f**) in Melzer’s, petri dish culture, (**g**) in lactophenol cotton blue, slide culture. Scale bars: 50 μm = (**c**,**d**); 10 μm = (**e**–**g**).

**Figure 16 jof-08-00877-f016:**
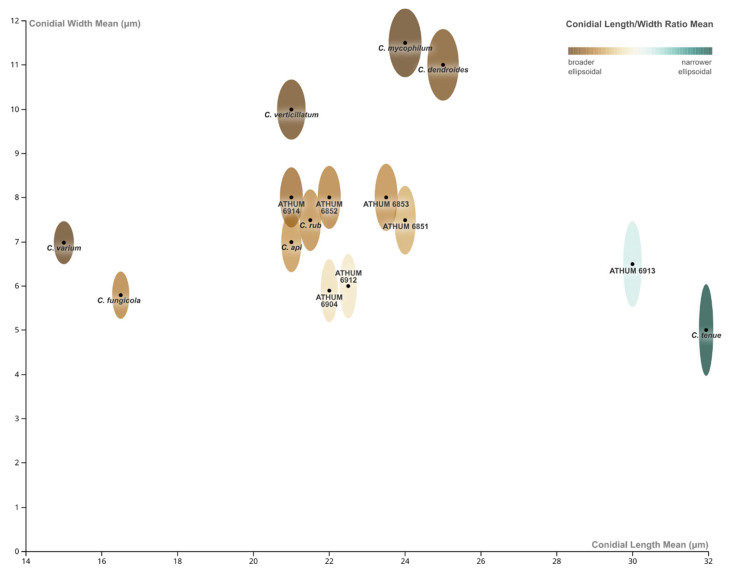
Customised bubble-type chart displaying three conidial dimensions (mean values of): conidial lengths (x-axis), widths (y-axis), and length/width ratios (Qm), represented by ellipsoidal shapes (not to scale), approximated by the corresponding mean length and width values, and coloured by an automated diverging colour scale as shown in the legend.

**Figure 17 jof-08-00877-f017:**
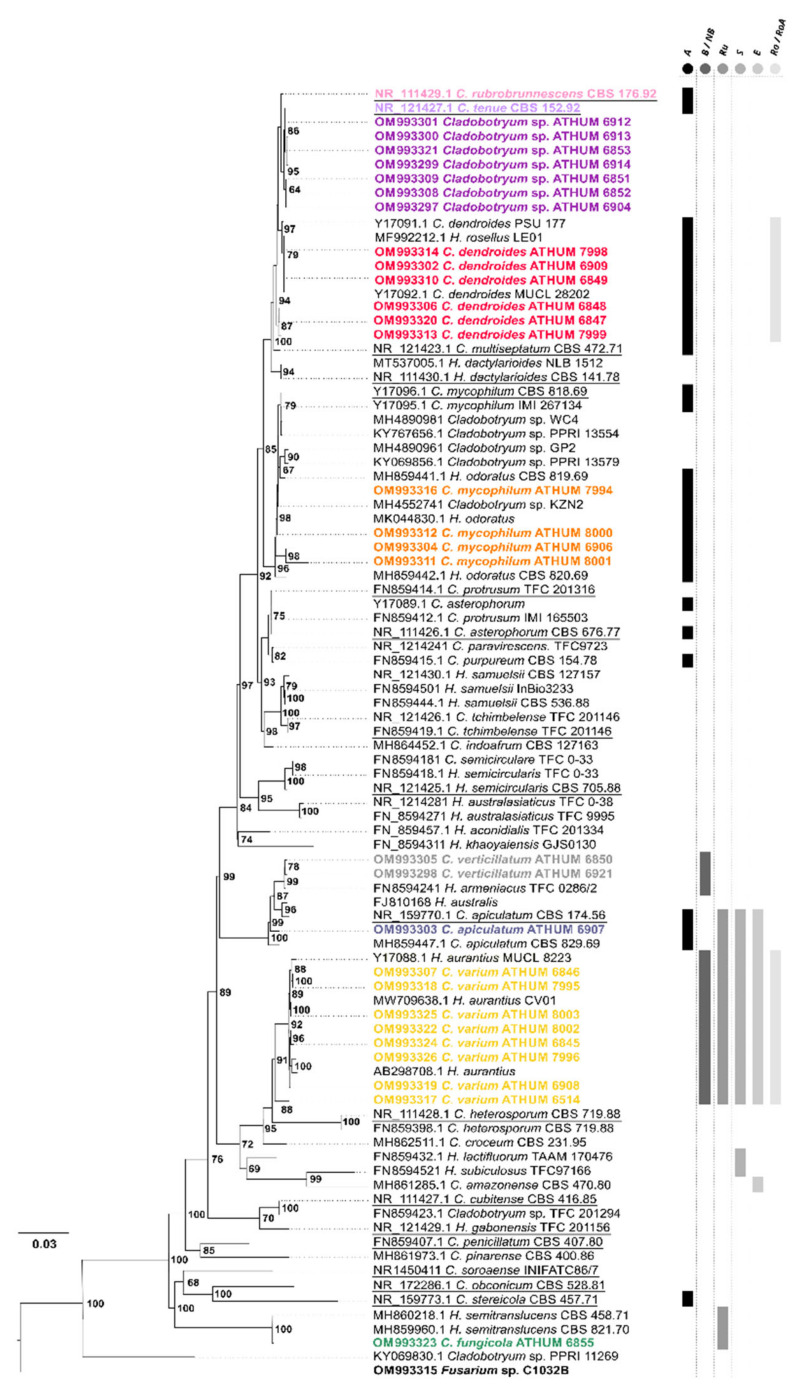
Phylogenetic tree of 92 *Cladobotryum*/*Hypomyces* strains as produced by Bayesian Inference based on ITS. *Fusarium* sp. C1032B (NCBI Acc. No. OM993315) was used as an outgroup. The NJ tree is identical. Phylogenetic relationships among taxa are mostly well supported. Posterior Probabilities (PP) are presented in black numbers. In nodes without numbers, PP values are lower than 70%. Greek strains described in this work were marked with bold and different colours. All type strains are underlined. According to the available literature [7,33,34,59,60,84,85,86,87,88,89,90], main pigmented metabolites and related compounds produced by the species *Cladobotryum*/*Hypomyces* are shown as bars on the right (A = aurofusarin; B/NB = bikaverin and norbikaverin; Ru = rugulosin; S: skyrin; E = emodin; Ro/RoA = rosellisin and rosellisin aldehyde).

**Figure 18 jof-08-00877-f018:**
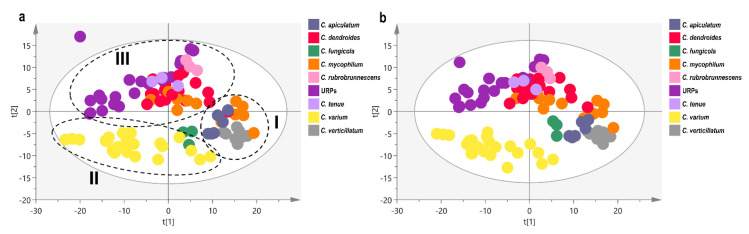
Score plot of the PCA (**a**) and PLS-DA (**b**) models of the hydroalcoholic extracts of all the examined *Cladobotryum* strains. In the PCA model, R2X = 0.25 and Q2 = 0.23 for the PC1 and 0.09 and 0.09, respectively, for the PC2. For the PLS-DA score plot axis t(1) has R2X = 0.25 and Q2 = 0.08 and t(2) has R2X = 0.09 and Q2 = 0.11.

**Figure 19 jof-08-00877-f019:**
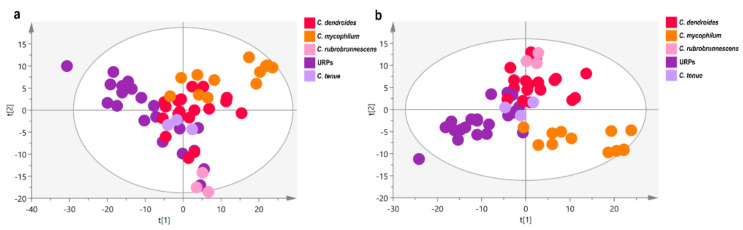
Score plot of the PCA (**a**) and PLS-DA (**b**) models of the hydroalcoholic extracts of the red-pigmented *Cladobotryum* strains. Axes quality parameters for PCA, PC1: R2X = 0.28 and Q2 = 0.24; PC2: R2X = 0.12, and Q2 = 0.12, and for PLS-DA, 1st component: R2X = 0.27, R2Y = 0.19 and Q2 = 0.18; 2nd component: R2X = 0.10, R2Y = 0.20 and Q2 = 0.11.

**Figure 20 jof-08-00877-f020:**
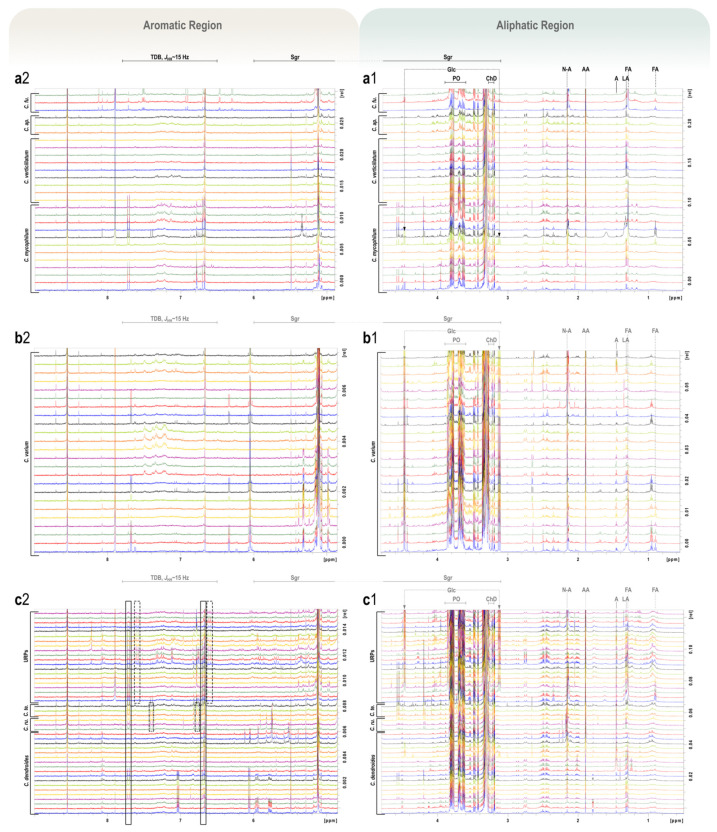
^1^H 1D NMR spectra of the hydroalcoholic extracts of the studied *Cladobotryum* strains showing the aliphatic (**1**) and aromatic (**2**) spectral regions of: (**a**) Cluster I; (**b**) Cluster II; and (**c**) Cluster III of PCA. TDB = *trans* double bond, Sgr = sugar moieties, Glc = glucose, PO = polyols, ChD = choline derivatives, N-A = *N*-acetylated compounds, AA = acetic acid, A = alanine, LA = lactic acid, FA = fatty acids. Each strain is represented in triplicates.

**Table 1 jof-08-00877-t001:** The strains of the genus *Cladobotryum* used in this study with reference to their fungal host, habitat, and geographic origin.

#	ATHUM Acc. Nos.	*Cladobotryum* Species	Fungal Host ^1^	Habitat, Geographic Origin
1	6907	*C. apiculatum*	*Russula* sp. (RUS, Rus)	*Fagus sylvatica* forest; Mt. Zigourolivado, Ag. Nikolaos, Karditsa
2	6847	*C. dendroides*	*Flammulina* sp. (AGA, Phy)	National Gardens, Athens, Attiki
3	6848	*C. dendroides*	*Polyporus varius* (POL, Pol)	*Fagus sylvatica* forest with *Juniperus communis*; Mt. Leivaditis, Xanthi
4	6849	*C. dendroides*	*Tricholoma* sp. (AGA, Tri)	*Abies cephalonica* forest; Mt. Parnitha, Attiki
5	6909	*C. dendroides*	polypore basidiome (POL)	*Fagus sylvatica* forest; Mt. Zigourolivado, Ag. Nikolaos, Karditsa
6	7998	*C. dendroides*	*Hohenbuehelia* sp. (AGA, Ple)	*Abies cephalonica* forest; Mt. Parnitha, Attiki
7	7999	*C. dendroides*	*Helvella lacunosa* (ASC)	*Platanus orientalis* forest with *Quercus* sp., *Abies cephalonica* and *Castanea sativa* (sporadically); Ag. Nikolaos, Eurytania
8	6855	*C. fungicola*	*Cortinarius* sp. (AGA, Cor)	*Quercus frainetto*, *Q. pubescens* and *Q. coccifera* forest with *Abies cephalonica* (sporadically); Mt. Oiti, Mesochori, Fthiotida
9	6906	*C. mycophilum*	*Hypholoma* sp. (AGA, Str)	*Castanea* sp. forest; Mt. Pilio, Magnisia
10	7994	*C. mycophilum*	*Inocybe* sp. (AGA, Ino)	*Abies borisii-regis* forest; Belοkomitis, Karditsa
11	8000	*C. mycophilum*	*Mycena* sp. (AGA, Myc)	*Platanus orientalis* forest with *Quercus* sp., *Abies cephalonica* and *Castanea sativa* (sporadically); Ag. Nikolaos, Eurytania
12	8001	*C. mycophilum*	*Agaricus bisporus* (cultivated) (AGA, Aga)	mushroom cultivation unit
13	CBS 176.92 ex-type	*C. rubrobrunnescens*	*Inocybe* sp. (AGA, Ino)	Regensburg, Germany
14	CBS 152.92 ex-type	*C. tenue*	agaric basidiome (AGA)	Keilberg at Regensburg, Germany
15	6514	*C. varium*	*Inocybe* sp. (AGA, Ino)	*Abies borisii-regis* forest; near Krikello, Eurytania
16	6845	*C. varium*	*Clitocybula familia* (AGA, Mar)	*Abies cephalonica* forest; Mt. Parnitha, Attiki
17	6846	*C. varium*	*Panellus* sp. (AGA, Myc)	*Abies cephalonica* forest; Mt. Parnitha, Attiki
18	6908	*C. varium*	agaric basidiome (AGA)	*Fagus sylvatica* forest; Mt. Zigourolivado, Ag. Nikolaos, Karditsa
19	7995	*C. varium*	polypore hymenophore (POL)	no data
20	7996	*C. varium*	polypore basidiome (POL)	*Picea abies*, *Pinus sylvestris* and *Betula* sp. Mt. Frakto, Rodopi
21	8002	*C. varium*	*Ganoderma* sp. (POL, Pol)	*Abies borisii-regis* forest with *Juniperus oxycedrus* (sporadically); Mt. Tymfristos, Eurytania
22	8003	*C. varium*	polypore basidiome (POL)	*Abies borisii-regis* forest; near Krikello, Eurytania
23	6850	*C. verticillatum*	*Lactarius subumbonatus* (RUS, Rus)	*Quercus* sp. forest; Mt. Lykaio, Megalopoli, Arkadia
24	6920	*C. verticillatum*	*Lactarius subumbonatus* (RUS, Rus)	*Quercus* sp. forest; Mt. Lykaio, Megalopoli, Arkadia
25	6921	*C. verticillatum*	*Lactarius subumbonatus* (RUS, Rus)	*Quercus* sp. forest; Mt. Lykaio, Megalopoli, Arkadia
26	6851	*Cladobotryum* sp.	*Laccaria laccata* (AGA, Hyd)	*Quercus frainetto*, *Q. pubescens* and *Q. coccifera* forest with *Abies cephalonica* (sporadically); Mt. Oiti, Mesochori, Fthiotida
27	6852	*Cladobotryum* sp.	*Lactarius salmonicolor* (RUS, Rus)	*Quercus frainetto*, *Q. pubescens* and *Q. coccifera* forest with *Abies cephalonica* (sporadically); Mt. Oiti, Mesochori, Fthiotida
28	6853	*Cladobotryum* sp.	*Russula* sp. (RUS, Rus)	*Quercus frainetto*, *Q. pubescens* and *Q. coccifera* forest with *Abies cephalonica* (sporadically); Mt. Oiti, Mesochori, Fthiotida
29	6904	*Cladobotryum* sp.	*Lactarius* sp. (RUS, Rus)	*Fagus sylvatica* forest; Mt. Zigourolivado, Ag. Nikolaos, Karditsa
30	6912	*Cladobotryum* sp.	*Tricholoma atrosquamosum* (AGA, Tri)	*Abies cephalonica* forest; Mt. Parnitha, Attiki
31	6913	*Cladobotryum* sp.	*Lactarius salmonicolor* (RUS, Rus)	*Abies cephalonica* forest; Mt. Parnitha, Attiki
32	6914	*Cladobotryum* sp.	*Cortinarius* sp. (AGA, Cor)	*Abies cephalonica* forest; Mt. Parnitha, Attiki

^1^ AGA = Agaricales; Aga = Agaricaceae; ASC = Ascomycota (Pezizales, Helvellaceae); BOL = Boletales; Cor = Cortinariaceae; Hyd = Hydnangiaceae; Ino = Inocybaceae; Mar = Marasmiaceae; Myc = Mycenaceae; Phy = Physalacriaceae; Ple = Pleurotaceae; POL = Polyporales; Pol = Polyporaceae; RUS = Russulales; Rus = Russulaceae; Str = Strophariaceae; Tri = Tricholomataceae.

**Table 2 jof-08-00877-t002:** The sequences of strains of the genera *Cladobotryum* and *Hypomyces* used in this study for the phylogenetic analyses; the newly-generated sequences of the Greek strains are given in black bold type.

#	Species	Isolate or Specimen Number	ITS GenBank Acc. Nos.	Country of Origin
1	*C. amazonense*	CBS 47080	MH861285.1	South Africa
2	*C. apiculatum*	CBS 829.69	MH859447.1	South Africa
3	*C. apiculatum*	CBS 174.56	NR_159770.1	Japan
4	** *C. apiculatum* **	**ATHUM6907**	**OM993303**	**Greece**
5	*C. asterophorum*	TFC 97-23	Y17089.1	Australia
6	*C. asterophorum*	CBS 676.77	NR_111426.1	Japan
7	*C. croceum*	CBS 23195	MH862511.1	South Africa
8	*C. cubitense*	CBS 416.85	NR_111427.1	Cuba
9	** *C. dendroides* **	**ATHUM6909**	**OM993302**	**Greece**
10	** *C. dendroides* **	**ATHUM6848**	**OM993306**	**Greece**
11	** *C. dendroides* **	**ATHUM6849**	**OM993310**	**Greece**
12	** *C. dendroides* **	**ATHUM7999**	**OM993313**	**Greece**
13	** *C. dendroides* **	**ATHUM7998**	**OM993314**	**Greece**
14	** *C. dendroides* **	**ATHUM6847**	**OM993320**	**Greece**
15	*C. dendroides*	PSU 177	Y17091.1	United States
16	*C. dendroides*	MUCL 28202	Y17092.1	Luxembourg
17	** *C. fungicola* **	**ATHUM6855**	**OM993323**	**Greece**
18	*C. heterosporum*	CBS 71988	FN859398.1	Cuba
19	*C. heterosporum*	CBS 719.88	NR_111428.1	Cuba
20	*C. indoafrum*	CBS 127163	MH864452.1	South Africa
21	*C. multiseptatum*	CBS 472.71	NR_121423.1	New Zealand
22	** *C. mycophilum* **	**ATHUM6906**	**OM993304**	**Greece**
23	** *C. mycophilum* **	**ATHUM8001**	**OM993311**	**Greece**
24	** *C. mycophilum* **	**ATHUM8000**	**OM993312**	**Greece**
25	** *C. mycophilum* **	**ATHUM7994**	**OM993316**	**Greece**
26	*C. mycophilum*	IMI 267134	Y17095.1	United Kingdom
27	*C. mycophilum*	CBS 818.69	Y17096.1	The Netherlands
28	*C. obconicum*	CBS 52881	MH861373.1	South Africa
29	*C. obconicum*	CBS 528.81	NR_172286.1	Belgium
30	*C. paravirescens*	TFC 97-23	NR_121424.1	Thailand
31	*C. penicillatum*	CBS 40780	FN859407.1	The Netherlands
32	*C. pinarense*	CBS 40086	MH861973.1	South Africa
33	*C. protrusum*	IMI 165503	FN859412.1	Zimbabwe
34	*C. protrusum*	TFC 201316	FN859414.1	Madagascar
35	*C. purpureum*	CBS 154.78	FN859415.1	USA
36	*C. rubrobrunnescens*	CBS 176.92	FN859416.1	Germany
37	*C. rubrobrunnescens*	CBS 176.92	NR_111429.1	Germany
38	*C. semicirculare*	TFC 03-3	FN859418.1	Sri Lanka
39	*C. soroaense*	INIFATC86/7	HE792978.1	Cuba
40	*C. soroaense*	INIFATC86/7	NR_145041.1	Cuba
41	*Cladobotryum* sp.	TFC 201294	FN859423.1	Madagascar
42	*Cladobotryum* sp.	PPRI 11269	KY069830.1	N/A
43	*Cladobotryum* sp.	PPRI 13579	KY069856.1	N/A
44	*Cladobotryum* sp.	PPRI 13554	KY767656.1	N/A
45	*Cladobotryum* sp.	KZN-2	MH455274.1	South Africa
46	*Cladobotryum* sp.	GP-2	MH489096.1	South Africa
47	*Cladobotryum* sp.	WC-4	MH489098.1	South Africa
48	** *Cladobotryum* ** **sp.**	**ATHUM6904**	**OM993297**	**Greece**
49	** *Cladobotryum* ** **sp.**	**ATHUM6914**	**OM993299**	**Greece**
50	** *Cladobotryum* ** **sp.**	**ATHUM6913**	**OM993300**	**Greece**
51	** *Cladobotryum* ** **sp.**	**ATHUM6912**	**OM993301**	**Greece**
52	** *Cladobotryum* ** **sp.**	**ATHUM6852**	**OM993308**	**Greece**
53	** *Cladobotryum* ** **sp.**	**ATHUM6851**	**OM993309**	**Greece**
54	** *Cladobotryum* ** **sp.**	**ATHUM6853**	**OM993321**	**Greece**
55	*C. stereicola*	CBS 457.71	NR_159773.1	Russia
56	*C. tchimbelense*	TFC 201146	NR_121426.1	Gabon
57	*C. tchimbelense*	TFC 201146	FN859419.1	Gabon
58	*C. tenue*	CBS 152.92	NR_121427.1	Germany
59	** *C. varium* **	**ATHUM6846**	**OM993307**	**Greece**
60	** *C. varium* **	**ATHUM6514**	**OM993317**	**Greece**
61	** *C. varium* **	**ATHUM7995**	**OM993318**	**Greece**
62	** *C. varium* **	**ATHUM6908**	**OM993319**	**Greece**
63	** *C. varium* **	**ATHUM8002**	**OM993322**	**Greece**
64	** *C. varium* **	**ATHUM6845**	**OM993324**	**Greece**
65	** *C. varium* **	**ATHUM8003**	**OM993325**	**Greece**
66	** *C. varium* **	**ATHUM7996**	**OM993326**	**Greece**
67	** *C. verticillatum* **	**ATHUM6921**	**OM993298**	**Greece**
68	** *C. verticillatum* **	**ATHUM6850**	**OM993305**	**Greece**
69	*H. australis*		FJ810168	China
70	*H. aconidialis*	TFC 201334	FN859457.1	Madagascar
71	*H. armeniacus*	TFC 02-86/2	FN859424.1	France
72	*H. aurantius*		AB298708.1	Japan
73	*H. aurantius*	CV01	MW709638.1	China
74	*H. aurantius*	MUCL 8223	Y17088.1	Canada
75	*H. australasiaticus*	TFC 03-8	NR_121428.1	Sri Lanka
76	*H. australasiaticus*	TFC 99-95	FN859427.1	Australia
77	*H. dactylarioides*	NLB 1512	MT537005.1	Australia
78	*H. dactylarioides*	CBS 141.78	NR_111430.1	New Zealand
79	*H. gabonensis*	TFC 201156	NR_121429.1	Gabon
80	*H. khaoyaiensis*	GJS01-304	FN859431.1	Thailand
81	*H. lactifluorum*	TAAM170476	FN859432.1	USA
82	*H. odoratus*	CBS 819.69	MH859441.1	The Netherlands
83	*H. odoratus*	CBS 820.69	MH859442.1	The Netherlands
84	*H. odoratus*		MK044830.1	China
85	*H. rosellus*	isolate LE01	MF992212.1	Spain
86	*H. samuelsii*	InBio3-233	FN859450.1	Costa Rica
87	*H. samuelsii*	CBS 536.88	FN859444.1	Cuba
88	*H. samuelsii*	CBS 127157	NR_121430.1	Puerto Rico
89	*H. semicircularis*	CBS 705.88	NR_121425.1	Cuba
90	*H. semitranslucens*	CBS 821.70	MH859960.1	Sweden
91	*H. semitranslucens*	CBS 458.71	MH860218.1	Russia
92	*H. subiculosus*	TFC 97-166	FN859452.1	Puerto Rico
93	*Fusarium* sp.	C1032B	OM993315	Greece

## Data Availability

The strains used in this study are deposited in the Culture Collection of the Mycetotheca ATHUM (NKUA), the sequences produced were deposited into GenBank (Acc. Nos. OM993297–OM993326) and the tree matrix is provided as Appendix A.

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
