# Peer review of "Polyphasic Systematics of the Fungicolous Genus Cladobotryum Based on Morphological, Molecular and Metabolomics Data"

_jof, 2022, doi:10.3390/jof8080877_

Round 1
Reviewer 1 Report
The Milic et al., study has focused on the systematics of the genus Cladobotryum through the combination of morphological, phylogenetic, and metabolomics data. In order to do so the authors have extensively worked with the multiple strains of the genus providing detailed descriptions of the morphological characteristics of the cultures. Furthermore, they have generated new ITS sequences, which they have combined and analyzed with other published data. Finally, they present detailed work on the metabolomics of these fungi and the contribution of the method in chemotaxonomy.
The manuscript is well written and I consider the study to be a significant contribution to our knowledge on the fungicolous genus Cladobotryum locally in Greece, but also in general.
Below I have a few comments that I think the authors need to consider, as well as some additional minor comments.
Main comments
What is the ITS percentage similarity between C. tenue, C. rubrobrunnescens and your unknown seven strains? Should these be considered one species? Obviously C. tenue has clear differences in the spore shape and size in comparison to the other strains that I mention here. However, I do think that the spore size and shape alone should not be strongly taken into consideration since for example you also have C. apiculatum being very similar to C. rubrobrunnescens, but at the same time these are phylogenetically distinct. I think this finding it is worthy a bit more attention in your discussion. What would be a good cut off value for species identification in Cladobotryum? I am not sure there is an easy answer to this question, but it is something to consider. A more thorough investigation of the clade these strains and species would be useful in the future.
While you discussed the advantages and limitations of the different approaches, I think it would be important to bring up what would be in your opinion the best way to study Cladobotryun in the future. Molecular data, metabolomics, morphology? Maybe a combination of the methods?
Lines 955-956: The statement that C. fungicola has an ancestral-like status both based on phylogenetic and metabolic data is not accurate in my opinion. It is the whole clade that contains C. fungicola that is ancestral not just the species. Furthermore, can you elaborate why this is supported based on metabolic data? My interpretation of figure 18 in combination with figure 17 would be that from a metabolic perspective cluster II seems to have more ancestral characteristics even though more species would be required to verify this.
Lines 956-958. The red pigmentation might not be only due to aurofusarin because the pigment has been also found in C. stereicola.
Lines 959-961: it would be better to avoid the statement as it is written now because based on the data you present the metabolite could have also appeared twice independently. Simply, we cannot extrapolate on the origin of the metabolite. One aspect you should bring up is that many times these metabolites are produced under various conditions and we cannot be sure whether they are present in other species as well. This could change completely the way your interpret these data here. One more point you should consider is that these metabolites are made by complex gene clusters and many times can undergo rapid evolution.
Minor comments
Figure 16: the resolution of the figure is not optimal. It would be nice if you had the full name of C. apiculatum and C. rubrobrunnescens fully written.
Table 2. Please also add that bold sequences are newly generated.
Line 452: I guess it should have been forest instead of for
Legend Fig 17, Line 796: Cladobotryum is misspelled
Legened Fig 17: Instead of species of interest you can better say species or clades discussed in the study.
Fig 17. The resolution of this figure is poor. The bold characters are not clearly visible and it is difficult to read the printed version.
Lines 959: “the” metabolite.
Reviewer 2 Report
There are two general issues (can be found also in highlights with comments in attached PDF):
I) Authors should use species binominals as currently solved and under the 1F1N rule. There is a reference, to my best knowledge - the newest one - Rossman et al. 2013 (cited in ms.!) where Hypomyces is conserved against the older Cladobotryum with valid arguments. If there is no newer and overweighting reference in this regards, the authors shoud
1) give another, strongly supported solution by themselves.
OR
2) use currently correct binominals - to my best knowledge there is two cases with combination in Hypomyces, some other could be done in the same manner in this very ms. according to established anamorph-teleomorph connections via phylogeny (IF teleomorph sequence used is reliably identified), and for the rest cases - use existing combination with Cladobotryum.
OR
3) Make combination with Hypomyces (relying on Rossman et al. 2013) if and where nobody has done that.
II) The second issue the authors should solve before publishing is related to poor figure quality. Especially the phylo-tree. All those problems are pointed out specifically in attached PDF as commented highlights.
In the end I wish to congratulate the authors on their polyphasic, integrative approach in taxonomy. I know that they invested huge amount of work in this and I will be very happy to see this manuscript published.

Reviewer 3 Report
I find this paper very interesting. Besides representing a milestone for further studies on taxonomy of Cladobotryum/Hypomyces, it introduces an intriguing perspective on the application of metabolomics in fungal taxonomy. My advice is that it can be published after minor revision, as per the below list of corrections/adjustments.
Lines 29 and 271: 'anamorph' not in italics;
lines 48-50: I suggest correcting to '...the ”one fungus, one name“ principle, a priority species name must be defined and established in current use for taxa having names for both morphs [13,14].';
lines 75-78: I suggest correcting to '...characterises this genus, in addition to overlapping of taxonomic characters among distinct species and the micromorphological variability which is often observed within a single species, make its taxonomy a challenging feat.';
line 120: use 'diversity' instead of 'mycodiversity' to avoid redundancy;
lines 142-144: delete 'growth medium'; indicate day/night hours; delete 'liquid culture';
lines 179-180: this version '...realise that our specimens range from a broader ellipsoidal shape on one extreme, to a narrower ellipsoidal shape on the other extreme' could be more fluent;
lines 271-272: change to '...; while the remaining 7 strains could not be identified at the species level';
line 280: 'Fthiotida' not in italics? The same for the other locations mentioned with reference to the strains examined?
line 426: 'similar' not in italics;
line 440: delete 'cell';
line 743: this sentence is too long; I suggest to divide as follows '...dendroides; all of them belong...';
lines 808, 822, 832, 834, 839, 841 and 858: correct to ‘score’;
lines 840-841: do not use article before percentages; change to ‘…described 76% of the variability, while predicting a lower share (51%).’;
line 925: delete ‘a cell of’;
line 944: add ‘rule’ or ‘principle’ after “one fungus, one name”;
line 949: delete ‘those of’;
line 951: correct ‘to’ to ‘for’;
line 953: correct to ‘strains’;
lines 959-962: as the authors state, lack of information prevents any inference concerning this metabolite. I suggest to modify this sentence as follows: ‘Additionally, bikaverin production only occurs in C. varium and C. verticillatum [7,37].’
Reviewer 4 Report
The manuscript Polyphasic Systematics of the Fungicolous Genus Cladobotryum Based on Morphological, Molecular and Metabolomics Data wants to identify Cladobotryum species by integrating morphological, molecular phylogenetic and metabolomic data into a polyphasic systematic approach, it is very meaningful for comprehensive study of the genus Cladobotryum. However, the seven isolates of Cladobotryum (ATHUM 6851, ATHUM 6852, ATHUM 6853, ATHUM 6904, ATHUM 6912, ATHUM 6913 and ATHUM 6914) finally didn’t determine their taxonomic position at species level by a polyphasic systematic approach in this study. Moreover, there are still many problems to be solved in the text, and some comments could be explored in the manuscript.
1. The writing of the manuscript needs to be greatly improved.
2. There are many formatting errors in this manuscript that need to be carefully revised, and I also marked a few in text.
3. Some species, such as Cladobotryum sp. ATHUM 6914 (line 718), have observed the sclerotioid structures, but their corresponding pictures of the structure didn’t display in figures, please add it.
4. The figure 17 and figure 20 should be replaced by figures with higher resolution ratio.
5. The text emphasize many times that different growth media and conditions have great impact on Cladobotryum species. Why not try more growth media and temperature for their isolated Cladobotryum species and describe their culture characteristics in detail?
6. The phylogenetic tree just based on ITS region didn’t distinguish well Cladobotryum species, and other genes, such as RPB1, RPB2 and TEF1 (reference Kadri Põldmaa, 2011), could be added in phylogenetic analysis.
7. There are also many errors in reference, and I marked a few in text. Please carefully examine all references.

Round 2
Reviewer 4 Report
The is an interesting study, and its aim is the identification of Greek strains of the fungicolous genus Cladobotryum, by integrating morphological, molecular phylogenetic and metabolomic data into a polyphasic systematic approach. However, morphological description of some Cladobotryum strain have errors, for example, the conidia of strain ATHUM 6913 should be descripted as 0-1 septate (line 729 is ‘mainly two-celled, but also one-celled’). For fungal identification, these structures described in text, such as conidiogenous cells and sclerotioid structures, should be displayed in plate. Photos showing conidiogenous cells or conidia should be clearer. The ITS-based phylogenetic tree didn’t distinguish different Cladobotryum species well. Now, multigene phylogenetic analysis is widely used for fungal identification. In this study, the polyphasic systematic approach just confirmed the morphologically identified Cladobotryum species, but didn’t determine their taxonomic position of these morphologically undentified Cladobotryum species at species level. So, this manuscript needs to be greatly improved in morphological study and phylogenetic analysis.
